# Pan-cancer characterisation of microRNA across cancer hallmarks reveals microRNA-mediated downregulation of tumour suppressors

Andrew Dhawan [1], Jacob G. Scott [2], Adrian L. Harris[1] & Francesca M. Buffa [1]

microRNAs are key regulators of the human transcriptome across a number of diverse biological processes, such as development, aging and cancer, where particular miRNAs have been identified as tumour suppressive and oncogenic. In this work, we elucidate, in a comprehensive manner, across 15 epithelial cancer types comprising 7316 clinical samples from the Cancer Genome Atlas, the association of miRNA expression and target regulation with the phenotypic hallmarks of cancer. Utilising penalised regression techniques to integrate transcriptomic, methylation and mutation data, we find evidence for a complex map of interactions underlying the relationship of miRNA regulation and the hallmarks of cancer. This highlighted high redundancy for the oncomiR-1 cluster of oncogenic miRNAs, in particular hsa-miR-17-5p. In addition, we reveal extensive miRNA regulation of tumour suppressor genes such as PTEN, FAT4 and CDK12, uncovering an alternative mechanism of repression in the absence of mutation, methylation or copy number changes.

[1] Department of Oncology, Medical Sciences Division, University of Oxford, Oxford OX3 7DQ, UK. [2] Translational Hematology and Radiology, Cleveland Clinic, Cleveland 44195, USA. Correspondence and requests for materials should be addressed to F.M.B. (email: francesca.buffa@oncology.ox.ac.uk)

The hallmarks of cancer outline the major phenotypic changes underlying the oncogenic process[1,2]. These changes characterise cancer as a disease, and may define actionable targets for therapeutic intervention. Since the definition of these characteristic hallmarks in 2001[1], and the subsequent genomic revolution that has occurred in the field of cancer biology, multiple groups have proposed gene expression signatures as biomarkers of these phenotypic hallmarks[3,4]. These signatures generally consist of a set of tens to several hundred coding genes, for which a summary metric of their collective expression is associated with a known hallmark, and may help with defining therapeutic strategies[5]. Encapsulated within this methodology and these signatures is a vast amount of biological discovery for particular genes implicated in the development and progression of these hallmarks. However, since the more recent publication of the updated hallmarks in 2011[2], there has been a second revolution in the field of genomics; namely, the discovery of the diverse, critical roles of non-coding RNAs in cancer.

Previously thought to be junk DNA, non-coding RNAs do not code for proteins, and consists of a diverse family of evolutionarily conserved species, including long non-coding RNAs (lncRNAs), circular RNAs (circRNAs) and microRNAs (miRNAs), among others[6,7]. Much effort has focused on the characterisation of non-coding RNAs, and early work has shown that these species, particularly miRNAs, are involved in a number of cellular developmental, and disease processes[8]. miRNAs exert their function primarily as repressors of protein production, functioning as post-transcriptional regulators of mRNA, inhibiting translation or encouraging transcript degradation. miRNAs exert their effects by complementary base-pair binding to a short 7–8 nucleotide seed region typically located on the 3′ untranslated region of the messenger RNA which they inhibit[6,9]. Whilst this complementary base-pair interaction defines many miRNA-target interactions, there is a class of non-canonical miRNA targeting that has been shown to occur throughout the transcriptome[10]. Such non-canonical interactions include many cases of imperfect seed matches, often with one mismatch, but remain difficult to predict[10]. A single miRNA is thought to able to exert its repressive effects on hundreds to thousands of transcripts, meaning that specific miRNAs may have very wide-ranging effects on cellular phenotype[6,11]. Despite this potential, due to the highly variable effect on the single target transcripts and the many factors involved in post-transcriptional gene regulation in addition to miRNA, the repressive signal on their target genes remains challenging to detect in clinical datasets. However, this is being abridged by the availability of large genomic datasets, and has been shown through the Cancer Genome Atlas project (TCGA)[12,13]. Large-scale studies for miRNA-mRNA interactions have begun to leverage the power of clinical datasets with thousands of patients, to detect small, context-specific effects[13,14]. For instance, Jacobsen et al. studied the miRNA-target interactions recurring across cancer types in the TCGA datasets[15]. This showed strong evidence for multiple miRNA concurrently regulating the DNA demethylation machinery of the cancer cell, through effectors such as TET1 and TDG, suggesting their important role in promoting cancer[15].

In addition to the difficulties of target prediction and small repressive effect sizes, a further complicating factor in the study of miRNAs is the relative promiscuity of their targets[16]. A given miRNA may have thousands of targets, with an increasing number experimentally verified, but often these targets possess significant differences in function[17]. This has led to an almost paradoxical finding about the effects of miRNAs, in that a single miRNA may theoretically exert effects in opposing directions within the cell[17]. This paradox is resolved by the observation that miRNAs likely play different roles depending on the environment in which they are expressed[16,18,19]. Therefore, in addition to the challenge of measuring the repressive effect of miRNAs within a transcriptome, the effect of a miRNA on a transcriptome may vary massively, depending on the relative abundance of each of its targets. This means that the effect of a miRNA on phenotype can only be observed in samples for which the transcriptomes are comparable in the expression of the key targets in consideration, and such effects are highly context-dependent. Recent work has aimed at generating an understanding of how competing miRNA targets regulate each other, and work, in particular by Chiu et al.[20] and by Xu et al.[21], has shown how these effects can be uncovered in a high-throughput manner.

Here, we show how the miRNA context-dependent action can be exploited to gain high confidence predictions in large clinical cohorts, uncovering known and unknown associations between miRNA and phenotype. Through the classification of tumour transcriptomes by gene expression signatures, we are able to generate hypotheses on the diverse roles of miRNAs in regulating the hallmarks of cancer. Our results point towards a scenario wherein the trancriptome of the cancer cell, known to be driven by dysregulation of tumour suppressor genes and oncogenes, is heavily regulated by miRNAs, extending the work by Jacobsen et al. and related studies[13–15]. We show that predicted miRNA-target associations that retain significance across multiple cancer types involve a number of critical tumour suppressor genes and oncogenes. Study of these tumour suppressor genes yields novel conclusions about their regulation, particularly with respect to their repression by miRNA, methylation and mutation, and the exclusivity of the occurrence of these modes of regulation across human cancers.

## Results

**Evaluation of Hallmark gene signatures across cancers**. We considered 24 previously well characterised gene signatures (Fig. 1 and Supplementary Note 2), and evaluated their performance on 15 well-annotated datasets with genomic and transcriptomic data, for a total of 7316 clinical samples[13,14]. We applied sigQC version 0.1.20, an R package to evaluate the basic statistical properties of gene signatures underlying their applicability across datasets[22]. We ran this package on all combinations of 15 datasets and 24 signatures, and tested the consistency of signature performance across cancer types, providing support to the application of the signatures to these specific datasets (Supplementary Note 2). In summary, each of the signatures considered, over the 15 epithelial cancer datasets, showed moderate to high expression, moderate to strong compactness, and moderate to high variability. This reassured us that the signal carried by these signatures was sufficiently strong and coherent to be unambiguously detected in these datasets, and sufficiently variable to differentiate between the clinical samples, such that we could proceed with further analysis

**Hallmark signatures associate with a complex miRNA network**. To determine the association between these gene signatures and the expression of all detectable miRNAs, we consider the summary score for each signature as the dependent variable of a linear model consisting of all miRNAs showing at least moderate univariate predictive ability for the summary score (Fig. 1a). Multivariable linear modelling with L1/L2 penalised regression was used to identify the miRNAs which showed the greatest predictive ability for each hallmark signature summary score across the cancer types considered, thereby identifying those miRNAs common to the gene signature across tumour types (see Methods and Fig. 1b). miRNAs were then ranked based on their final model coefficient (reflective of the strength of association to the

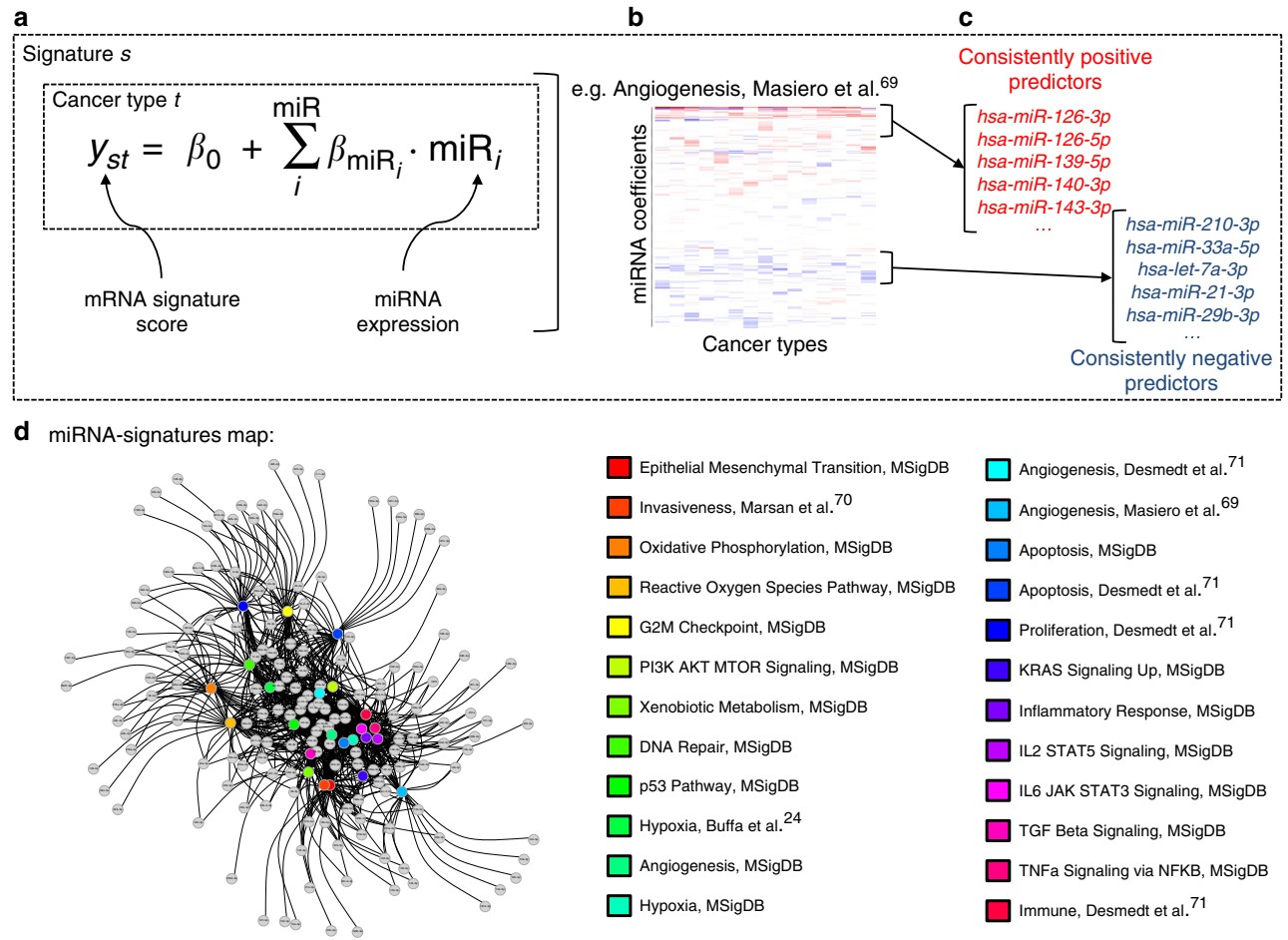

**Fig. 1** Overview of approach used to identify hallmarks-associated miRNA. **a** Overview of the linear model used in the fitting, for each gene signature and cancer type under consideration. **b** Example of a heatmap depicting the values of the coefficients identified for the miRNA predictors (rows), across cancer types (columns) for our previously developed angiogenesis signature[69]. **c** Consistently positive and negatively ranking miRNA coefficients, identified as statistically significant by the rank product statistic, are taken as the positive and negative hallmark-associated miRNA for each hallmark signature. **d** Network map of signatures (coloured circles) and their positively associated miRNA (grey circles), connected by edges when an association was found, highlighting strong interconnectivity between distinct molecular signatures

signature), and miRNAs consistently ranking highly as positive predictors of a given hallmark signature across cancer types were aggregated, from which statistically significant miRNAs were isolated as signature-associated miRNAs using the rank product statistic, as depicted in Fig. 1c. Likewise, for each gene signature, the miRNAs most consistently ranked as strong negative predictors of signature score across cancer types were aggregated by the same rank-based methodology, as depicted in Supplementary Figure 11 (negatively-associated miRNA). This analysis reveals both many known and unknown significant associations between miRNA and gene signature scores, facilitating an understanding of the miRNA involved with hallmark phenotypes, providing both novel hypotheses, and adding to evidence for existing ones.

To verify the validity of our predictions, we considered the example case of miRNAs found to associate significantly with the hypoxia signatures considered. Hypoxia is one of the most studied microenvironmental perturbations in the context of miRNA regulation, and one with a very well-defined pathway, controlled largely by a single transcription factor, HIF-1α[23]. Taking the intersection of the sets of miRNAs found to associate positively with the two previously validated hypoxia gene signatures (Hypoxia, Buffa et al.[24] and Hypoxia, MSigDb[25]), we obtained high confidence predictions for hypoxia-associated miRNAs.

As shown in the Tables associated with Supplementary Note 3, this analysis reveals that many of the miRNAs found to be commonly associated with both hypoxia gene signatures have been previously identified as hypoxia regulated. High confidence predictions are made for: hsa-miR-210-3p[26], hsa-miR-21-3p, hsa-miR-21-5p, hsa-miR-23a-5p, hsa-miR-23a-3p, hsa-miR-24-3p, hsa-miR24-2-5p, hsa-miR-27a-5p[27], let-7e-5p, let-7e-3p[28], let-22-5p, let-22-3p[29]. This analysis also suggests significant, pan-cancer, potential roles for other members of the let-7 family of miRNAs in hypoxia; namely, let-7b-5p, let-7b-3p, let-7d-5p, let-7d-3p, as well as hsa-miR-223-3p, hsa-miR-18a-5p and hsa-miR-28-3p, which have potentially escaped the notice of other approaches.

In the context of all gene signatures considered, we identify a global underlying map connecting each miRNA to each predicted associated gene signature. As shown in Fig. 1d, this is a highly interconnected and complex network, with the conservation of a core set of miRNAs shared across the hallmarks of cancer. A similar analysis reveals an analogous result for the miRNA-hallmarks network for the miRNAs negatively associated with the signatures, as described in Supplementary Note 4. To validate the reproducibility of these results, we rebuilt the signature-miRNA linear model using a large independent dataset, the Metabric breast cancer cohort[30]. The miRNA identified as positively and

negatively associated with the hallmarks gene signatures in this dataset showed highly significant concordance, over the majority of signatures, with the results obtained in analysis of the TCGA BRCA dataset (Supplementary Figure 12, Supplementary Note 5a). Moreover, although we validate on a breast cancer dataset, we have also been careful to show that our results are not overfit to breast cancers. Repeating our analysis in identifying miRNA without breast cancers included yields a very strong overlap, at least 75% in each signature, and above 80–90% in most cases, as we report in Supplementary Note 5b, and Supplementary Figure 13.

**miRNA family members show opposing oncogenic behaviours.** Subsets of miRNAs that share common, evolutionarily-conserved sequences or functional motifs in their sequences are typically organised into miRNA families[31,32]. Our analysis revealed that in a number of cases, miRNAs from the same families were significantly over-expressed and others significantly under-expressed in association with each of the gene signatures of the hallmarks of cancer, and this was consistent across cancer types (Supplementary Note 6, Supplementary Figure 16). For example, the miR-17/17-5p/20ab/20b-5p/93/106ab/427/518a-3p/519 and let-7/98/4458/4500 families have multiple members across

signatures both in statistically significant positive and negative associations.

This highlights once more the context-dependent nature of miRNA regulation, and the potentially antagonistic behaviours of miRNAs from the same family, supporting findings from ours and collaborators' work[33,34]. This suggests the need for additional context-dependent functional miRNA classifications uncovering key functional associations while complementing the current sequence and motif-based classifications.

**Hallmarks-associated miRNA targets are enriched for TSG.** Starting from the positively associated miRNAs with each hallmark gene signature, we aimed to identify which predicted miRNA-target pairs showed strong evidence of negative regulation across cancer types. This would confirm that the miRNA is functional, and would reveal which targets is acting upon in the specific context. We considered for this a comprehensive list of predicted targets provided by the union of five miRNA target prediction algorithms, as implemented by the package miRNAtap version 1.14.0[35] (see Methods section). Thus, we considered miRNA and predicted target mRNA pairs for which there was a statistically significant negative Spearman correlation of expression across at least five cancer types, hence evidence supporting

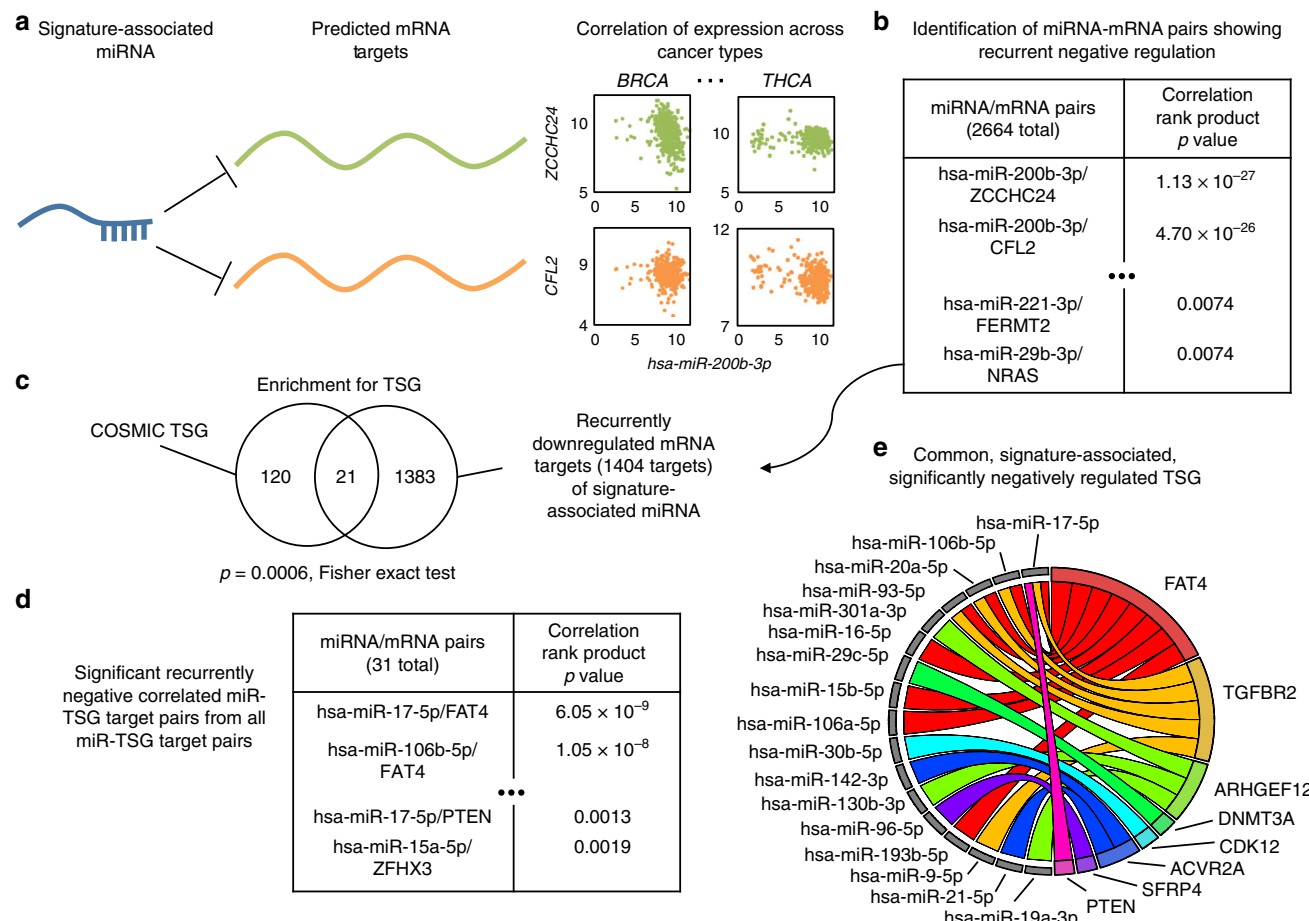

**Fig. 2** Approach used for interpreting miRNA-target interactions. **a** First, miRNA-target pairs for each positively associated hallmark-associated miRNA were identified, and the correlation between these was determined. **b** Next, the correlations across cancer types were aggregated, and those identified as consistently negative-ranking were identified with the rank product statistic. **c** Among this list of miRNA-mRNA target pairs, there was highly significant enrichment for tumour suppressor genes, as identified by the Fisher exact test. **d** The same procedure as described in **a** and **b** was repeated for all miRNA and all predicted target TSG pairs, with each TSG considered individually. **e** From the lists identified in **b** and **d**, we identified those miRNA-TSG pairs in common, and plot their interactions on a circos plot, showing the repressive actions of each miRNA on its predicted target TSG

active miRNA regulation. Then we used a rank-product test to identify the miRNA-target pairs showing consistency across cancer types (Fig. 2a). As depicted by the process in Fig. 2b, c, analysis of these significant miRNA-target pairs revealed a strong enrichment for tumour suppressor genes (as defined by the COSMIC database list of 141 TSG), as might be expected for miRNAs associated with oncogenic processes ($p = 0.0006$, two-sided Fisher's exact test). To further test the significance of increased number of TSG repressed by the signature-associated miRNAs, a bootstrap resampling-based approach (see Methods section), was devised. From all expressed miRNAs across cancer types that could have been chosen as signature-associated miR-NAs, random lists of the same length as the number of signature-associated miRNAs were chosen, and, via an analogous approach as above, the number of repressed TSG for these miRNAs was determined. Repeating this resampling 1000×, the probability that 21 or more TSG were repressed by the chosen miRNA was $p = 0.017$ (empirically determined), again suggesting strong significance in the enrichment for TSG among the repressed targets of the hallmarks signature-associated miRNA. This suggests that miRNA-mediated repression of tumour suppressor genes may be relatively common, significant, and associated with the phenotypic hallmarks of cancer.

A different picture emerged upon repeating this analysis for oncogenes, and for the miRNAs found to be significantly negatively associated with one or more hallmark signatures. We identified 1283 significantly anti-correlated miRNA-target pairs for these downregulated hallmark-associated miRNAs. Likewise, analysing all predicted miRNA-oncogene interactions among the 231 COSMIC oncogenes, there were only 2 showing significant anticorrelation across tumour types with their predicted target miRNA (ESR1 and ABL2). Taking the intersection of these lists of 2 COSMIC oncogenes and the 1283 miRNA-oncogene pairs associated with gene signatures identified only ESR1 (interacting with miR-18a-5p and miR-130b-3p) in common ($p = 1.2 \times 10^{-5}$, Fisher's exact test). This suggests that ESR1, oestrogen receptor alpha, may play a significant role across the hallmarks of cancer, and de-repression by reduction of its miRNA-mediated repression may play a role in cancer phenotype, and ultimately, oncogenesis[36,37]. Importantly, this result is also a strong negative control for our analysis, and it concurs in supporting the common oncogenic role of miRNAs via co-ordinated repression of tumour suppressor genes.

**A core set of TSG associate with hallmark signatures**. Next, we asked whether our results could be biased by the initial selection of miRNAs, namely the ones associated with the cancer hallmarks. To answer this, we conducted a complementary analysis. We sought to determine which of the miRNA-mediated tumour suppressor genes showed significance in downregulation, in the context of all other tumour suppressor genes. Thus, we repeated the previous analysis extended to all predicted miRNA-TSG pairs, individually for each TSG, considering again the significant associations across at least five cancer types, and then collated with a rank product test (Fig. 2d). This second part of our analysis provided the miRNAs associated to each TSG individually that showed strong significance across cancer types, to mitigate the bias accumulated by comparing regulation of multiple TSG in the same analysis (flow diagram shown in Supplementary Figure 1). Considering the miRNA-TSG pairs found to be of significance in both analyses from Fig. 2c, d, we identified a set of 22 miRNA-TSG pairs, comprising 8 TSG (FAT4, TGFBR2, ARHGEF12, DNMT3A, CDK12, ACVR2A, SFRP4 and PTEN) and 17 miR-NAs in Fig. 2e, in common. We show also that the miRNAs associated to each of these TSG are expressed at significantly

higher levels in wildtype cases for the associated TSG, across multiple cancer types (Supplementary Figure 17, Supplementary Note 7). Taken together these results demonstrate that for these tumour suppressor genes, (i) miRNA-TSG interactions are significantly enriched for across cancer types, (ii) miRNA-TSG interactions are strongly associated with the phenotypic hallmarks of cancer, and (iii) miRNA-TSG interactions may show increased importance in cases with wild-type TSG. Importantly, the conserved miRNA-TSG regulation across cancer types reveals this as a potential global mechanism, alternative to genetic mutations, to achieve functional inhibition of TSGs in cancer.

**Hallmarks-associated miRNAs show context-dependent action**. To further understand if the above presented miRNA-target associations were cancer-specific, we sought to determine whether similar conclusions could be reached when analysing non-tumour tissues. Starting from the associated adjacent normal tissue datasets from TCGA for tissue types with at least 20 samples for both miRNA and mRNA expression (BRCA, UCEC, HNSC, KIRC, LUAD and BLCA), we fitted a linear model for gene signature score as a function of all miRNA, for each signature, in each of the six tissue types. Aggregating coefficients across tissue types, we found that, while a highly significant number of miR-NAs associated with the gene signature scores across tissue types are the same as uncovered for the cancer tissues, there are significant differences. Across signatures, an overlap of on average 54% was observed for signature-associated miRNAs, showing high statistical significance for miRNAs positively and negatively associated with signatures ($p < 10^{-19}$ in all cases, by Fisher's exact test). We note that fewer normal samples were considered in this analysis as compared to the previous analysis in cancer samples, due to more limited availability of matched-normal tissues. Thus, we first ensured the validity of all gene signatures considered on normal datasets using sigQC; this showed that normal datasets are comparable in quality for the application of these gene signatures to tumour datasets (Supplementary Figures 2–10). Further, in Supplementary Note 5c, we describe the effects of using a reduced set of tumour samples the same size as the normal samples we considered; this showed a similar degree of overlap in signature-associated miRNAs, depicted in Supplementary Figure 14.

Examining the targets of these positively signature-associated miRNA from normal tissues, we identified 233 recurrently negatively correlated miRNA-target pairs, of which two contain miRNA-TSG pairs (CEBPA and NCOA4). However, this overlap of the 142 unique genes among the 233 miRNA-target pairs with the 141 COSMIC tumour suppressor genes does not show significance, and may be due to chance alone ($p = 0.26$ by Fisher's exact test). Thus, while the biology captured by the phenotypes of the gene signatures may be consistent, more than chance alone would predict, between tumour and normal samples, the resultant miRNA-target interactions are significantly different, and miRNA-TSG enrichment is not retained among normal tissue samples, highlighting the strong context dependency of these associations. Moreover, when considering a reduced set of tumour samples, to account for the smaller number of normal tissue samples, we are still able to capture these differences in miRNA phenotypes, as reported in Supplementary Note 5c.

Finally, we investigated whether the context-dependence of miRNA associations could also be detected in different breast cancer subtypes. Comparing the miRNA identified as hallmarks-associated between the basal and luminal B subtypes of breast cancer in the TCGA dataset, chosen for their relatively large sample sizes, we show that there are indeed significant differences

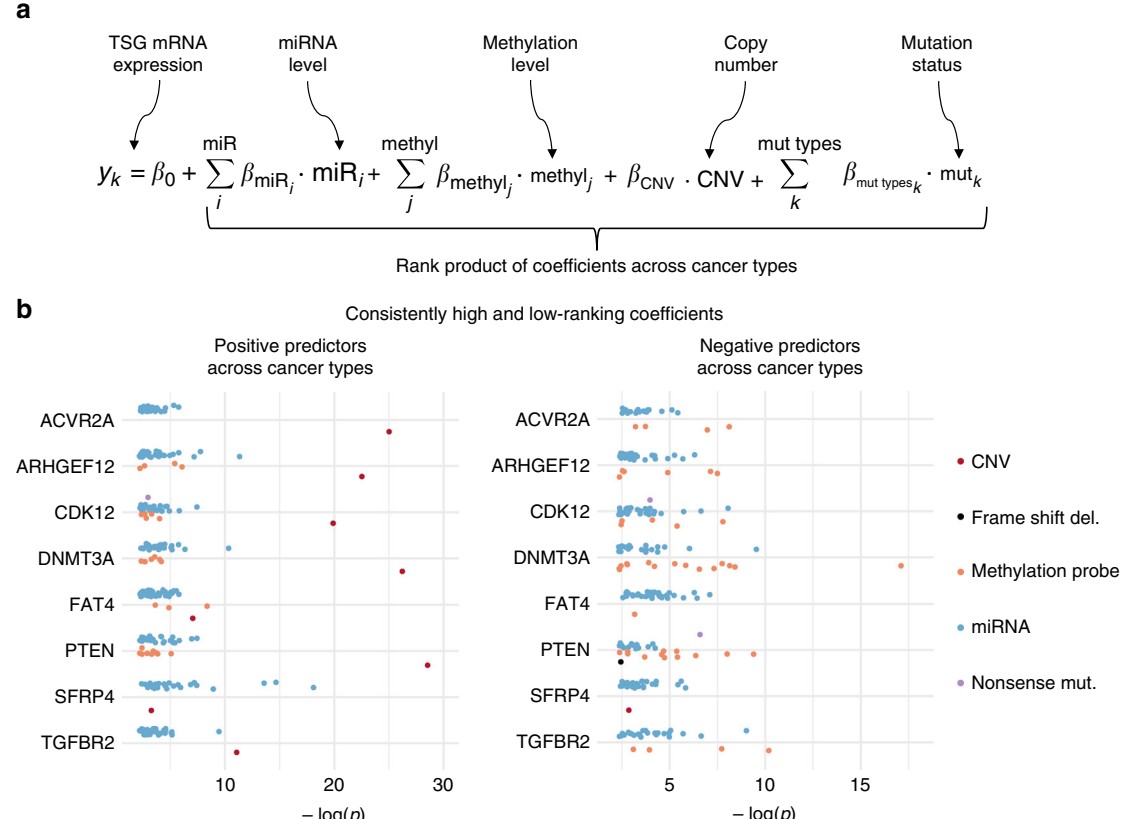

**Fig. 3** Approach used in determining the regulation of each TSG identified as potentially significantly miRNA-regulated. **a** The linear model used whilst determining predictors of TSG mRNA expression. **b** Model coefficients were aggregated across cancer types with the rank product statistic, and those identified as statistically significant positive and negative predictors are depicted alongside the -log of their rank product $p$-value

among key miRNA associated to the hallmarks signatures for these two breast cancer subtypes. These results are described in Supplementary Note 5d, and depicted graphically in Supplementary Figure 15.

**Copy number and mutational status determine TSG expression.** We next broadened our analysis and sought to characterise more generally the determinants of the expression of TSG not limited to miRNA. In particular, we consider an approach integrating multiple lines of genomic information; in addition to the miRNA expression, we considered methylation status, copy number and mutational status of the TSG (see Methods section and Fig. 3a). Notably, when considering the impact of miRNA expression in this model, we considered all reported miRNAs to potentially discover novel miRNA-target interactions instead of limiting the model to predicted miRNA-targets pairs. We then fit this model with penalised linear regression over the various cancer types (see Methods section), and then subsequently aggregated coefficients by the rank product statistic to identify recurrently positive and negative predictors across cancer types, for each of the 8 tumour suppressor genes identified in Fig. 2e. This analysis yields both expected results, such as the important predictive role of copy number for each of the tumour suppressor genes, as seen in the left panel of Fig. 3b, and novel associations, such as the positive association of many miRNAs, and some methylation probes with TSG expression in some cases. Positively-associated miRNAs arose in this analysis as a result of the inclusion of all miRNAs expressed in each cancer type, as opposed to those only predicted to target the TSG, so that novel associations could be uncovered. The positively associated

miRNA may appear to be co-expressed for a variety of reasons, such as competitive endogenous mRNA (ceRNA) interactions, downregulation of repressors of the TSG, or presence on a nearby genomic locus subject to the same enhancer or promoter. We do note, however, that our approach in using penalised linear regression works to minimise the effects of miRNAs present on a nearby genomic locus, as copy number has been included as a covariate in the linear model. That is, penalised linear regression functions by adding two penalty terms in the linear model—an L1 penalty reduces the overall number of predictors, and an L2 penalty helps to distribute coefficients' values for correlated covariates, such as copy number and miRNA expression on the same locus.

Likewise, the identified modes of negative regulation give expected results, with non-sense mutations and frame shift deletions consistently negatively associated with TSG mRNA expression. Further, because this analysis was done with all miRNAs, and not just those predicted to have a given TSG target, these results may reveal novel miRNA-TSG interactions not covered by current prediction algorithms, and represented by the negatively correlated miRNA with each TSG. The complete rank product tables and all autocorrelation matrices can be found in the Supplementary Files. In addition, the listing of all probe sets used and mutation types considered in this analysis for each TSG is listed in Supplementary Note 8, Supplementary Tables 1 and 2.

**TSG expression regulated either by miRNA or methylation.** Once the modes of regulation and their relative importance was established (Fig. 3), we sought to determine the relative occurrence of each of these modes of regulation. We identified which

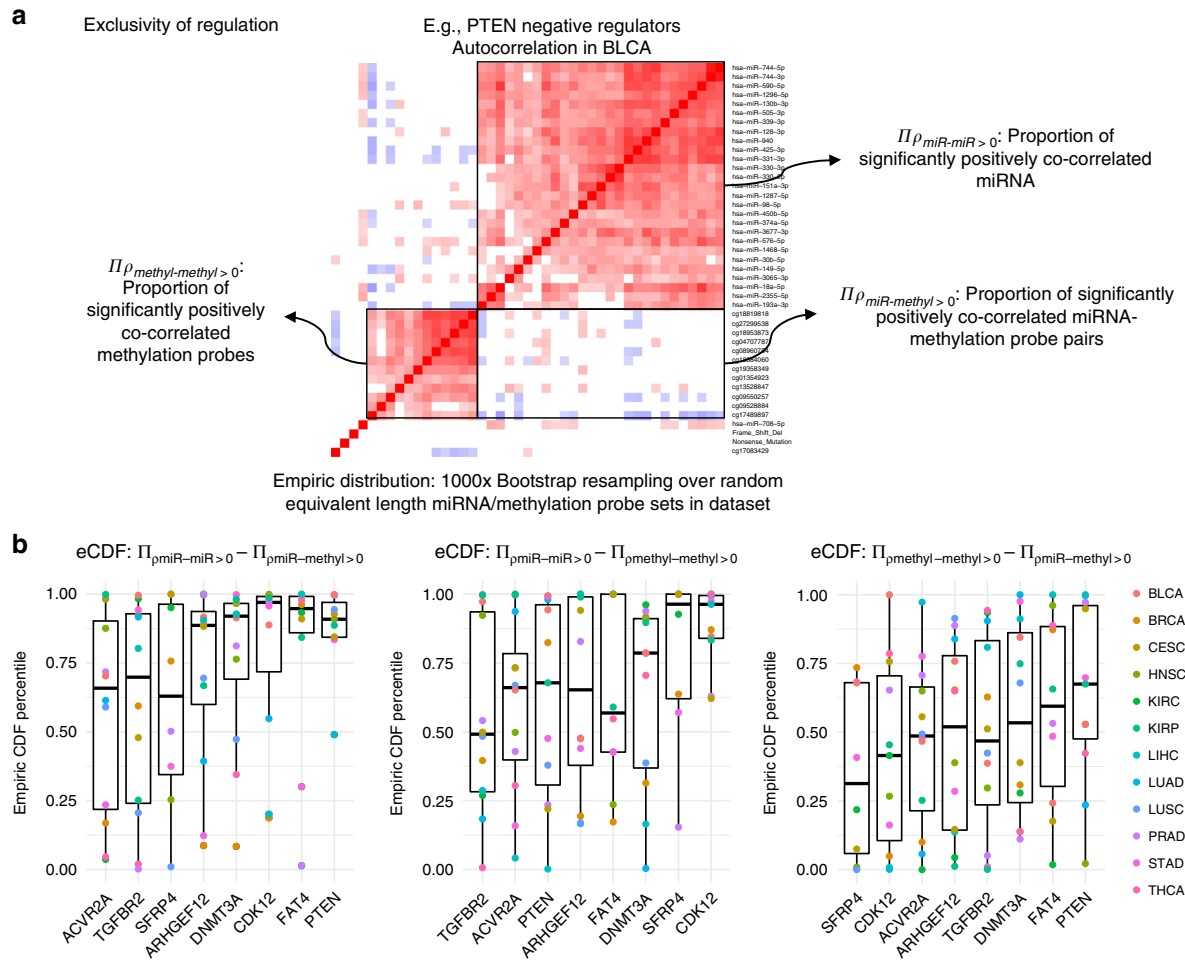

**Fig. 4** The approach used to determine the exclusivity of each mode of gene regulation on expression for the TSG considered. **a** Depiction of the autocorrelation heatmap for the expression of the various negative regulators of the tumour suppressor gene, and the variables considered and their meaning, as depicted. **b** Plots depicting the spread of the percentiles on the empiric cumulative density function (CDF) for the distributions for the pairwise differences of the variables identified in **a** through a bootstrapping-based analysis, as described in the Methods section. Centre line of boxplots depicts median, bounded by interquartile range (IQR), and whiskers extending to 1.5 times the IQR

negative regulators co-occurred with each other, and conversely which were exclusive repressors (Fig. 4a). A cursory analysis of autocorrelation heatmaps (e.g., Figure 4a) revealed that in some cases, the regulation by miRNA appeared to be exclusive from the regulation by methylation probes; methods are described in detail in Supplementary Note 8 and Supplementary Figure 18. A full series of heatmaps for all cancer types considered and all tumour suppressor genes with their associated negative regulators identified is presented in Supplementary Note 9, Supplementary Figures 19–26, and for an independent dataset in Supplementary Figure 27, details described in Methods. These results suggest that TSG expression can be altered by either miRNA or methylation, in addition to deletion or mutation, in a BRCA-ness-like phenomenon[38]. To characterise this, we devised a bootstrap resampling based approach (see Methods section), to determine significance of the difference in co-correlation between the miRNA and the methylation probes themselves, and then with each other. For each cancer type, we calculated the significance value of this proportion (Fig. 4b), and from this analysis, it arose that for each of the TSG considered, there are tumour types in which the regulation is consistently exclusive. Further, across multiple cancer types, three key tumour suppressor genes, PTEN, FAT4 and CDK12, consistently tended towards exclusivity in their regulation, lending support for the importance of miRNA-

based regulation of these genes. We further used these negatively associated miRNA and methylation probes, along with mutation status, to define subgroups of samples, for which we show decreased TSG expression in the subgroups with high expression of these miRNA or high methylation of these probes, in Supplementary Figures 28–37 in Supplementary Note 10. We also show that the miRNA-high and highly methylated samples have transcriptomes altered in a similar manner as in TSG mutated cases, via an analysis of differentially expressed genes in both cases, with significantly positively associated fold changes across cases, in Supplementary Figures 38–46 in Supplementary Note 10.

**ARHGEF12, SFRP4, TGFBR2 and cognate miRNA define BRCA subtype.** Next, we sought to identify associations with tumour molecular subtypes, and as an initial analysis chose the molecular subtypes of breast cancer, owing to both the well-defined subtypes and the relatively large number of cases available for each subtype. An analysis of the eight identified tumour suppressor genes consistently negatively downregulated by miRNA across cancer types shows that in many cases, their mRNA levels are associated with breast cancer molecular subtype. In particular, the basal subtype shows the lowest median expression of ARHGEF12, SFRP4 and TGFBR2, as compared to

normal tissue, luminal A, B, Her2 amplified, or normal subtypes of breast cancer as shown in Supplementary Figure 47 in Supplementary Note 11, and this association is retained when cases are restricted to wildtype expression of ARHGEF12, SFRP4, and TGFBR2. At the level of the associated miRNAs identified as negative regulators of these TSG, we show that the median expression of these miRNA is also significantly associated with breast cancer molecular subtype, and this association is reversed when related to TSG mRNA expression, as expected. We have also shown that these associations are preserved when samples with non-silent mutations in the TSG are removed. For further validation, we also show reproducibility of these TSG and miRNA associations to breast cancer subtype in the independent Metabric dataset ($N = 1293$)[30].

## Discussion

In this work we have carried out a comprehensive and rigorous association analysis of human transcriptomic and genomic data to leverage an understanding of the role of miRNA in regulating complex phenotypes, through the lens of established gene expression signatures of cancer hallmarks. Gene signatures represent transcriptomic association and we utilised them in two key ways, adding significant power to the analysis. Firstly, we use gene signatures to understand the relationship between non-coding RNA and a given cancer phenotype; this exploits the phenotypic associations intrinsic to established gene signatures. Secondly, because miRNA can only repress mRNA that are present in sufficient quantity in a cell, when inferring function, it is vital to group transcriptomic profiles by miRNA targeted gene expression. This allows for an understanding of the context-dependent miRNA-mediated gene regulation important to the phenotype one wishes to uncover. Thus, this analysis represents a novel and powerful assessment of the intricacy of miRNA regulation of phenotypes, which is particularly important in the context of a complex disease, such as cancer.

Our work begins with ensuring applicability of the gene signatures, and then for each signature, we gain an understanding of the miRNA both significantly upregulated and downregulated in association with the signature summary score. From this, we obtain the network shown in Fig. 1, which describes in a detailed fashion across cancer types, the contribution of individual miRNA to cancer phenotypes. We also show reproducibility of this network in an independent dataset, by considering the overlap with the network reconstructed using the Metabric dataset and the same gene signatures. Moreover, repeating this analysis by grouping the miRNAs significantly upregulated and downregulated by miRNA family, illustrates that many miRNA families participate with members acting antagonistically across the hallmarks of cancer; including 4 of the top 5 most common miRNA families in our analysis (miR-25 family, miR-17 family, miR-15abc family and let-7 family). By virtue of the high similarity between their seed regions, often resulting from common evolutionary ancestry, miRNA families are thought to consist of miRNAs with similar biological function, and to a certain extent redundantly targeting the same mRNAs[31,39]. Our results challenge the prevailing hypothesis of miRNA families acting in a generally coordinated fashion across multiple phenotypic states, and highlights the frequent context dependent behaviour of individual miRNA themselves, regardless of grouping by family[31,32]. Further strengthening the argument for context-dependent actions of miRNA is the observation that we have made for the gene signature network reconstructed from six tissue types with samples of adjacent normal, non-tumour tissue. While a significant proportion (54%) of miRNA found to be associated with the gene signatures are the same as for the tumour

tissues, which can be expected due to the common tissue origin, the analysis of the targets of these miRNA reveals that they do not show enrichment for TSG in the normal tissue, again highlighting the context dependency in miRNA-mediated gene regulation. We note that this analysis of context dependence could have been repeated using different subgroups of cancers, or tumour sub-types, instead of tumour samples versus normal samples. However, in our work, we sought to retain as much statistical power as possible, and confidence in our analysis, by comparing large groups of tumours in multiple datasets with meaningful distinction to reduce the impacts of noise and of model overfitting, but future studies with larger cohorts may hone in on these differences.

Related to the discussion of statistical power, is the discussion of the validity of the miRNA-target interactions inferred from various target prediction algorithms. In this study, miRNA-target gene interactions are predicted using the miRNAtap database in R, version 1.14.0, as described in the Methods section. While we have not repeated our analysis further to include greater stringency in the target selection (e.g., requiring a miRNA-target interaction predicted by more than two independent sources to be considered), the analysis methodology itself was designed to obtain high-confidence targets. Initially, we retained many possible miRNA/mRNA targets, with a balance of potential false-positives, requiring a minimum of two sources predicting the interaction. In this way, we sought to include a reasonable number of interactions, while not restricting to those which are predicted by the commonalities of each algorithm. With this more comprehensive list, we tested directly which of these miRNA/target pairs themselves show repressive potential using correlations in the data, and then further refine this list using the rank product statistic. Thus, for the analysis we present, we move from a reasonably wide and comprehensive list of potential targets initially, and refine these using the power afforded by the large datasets used.

As might be expected, given the complexity of the action of non-coding RNA, we show in this work that for a given phenotype, single miRNA-target interactions do not account for observed behaviour; rather it appears to be that these changes come from an expressed network of miRNAs, interacting with a set of targets in a coordinated manner, tuning the transcriptome to achieve the complex phenotypes of cancer. That is, because the targets of a given miRNA are predicted to be variable in their function, and are not all present in every sample at repressible concentrations (often necessitating the use of an expression level filter in our analysis), the same miRNA can be associated with opposing phenotypic effects in different contexts, as reported by Denzler et al.[40] for competing endogenous mRNA (ceRNA). ceRNA have also been identified in a high-throughput fashion by Chiu et al.[20] and Xu et al.[21] and an approach to their identification, describing the necessary experimental and statistical pre-requisites has been reported by Smillie et al.[41], with recent work identifying such networks involving PTEN, for instance[42]. We show that the behaviour of miRNAs is highly context dependent, and through our pan-cancer analysis, we have aimed to reduce the complexity of this context dependency by only selecting those interactions significantly occurring across cancer types. However, we caution that because miRNAs are so context dependent, sample purity arises as an important issue in identifying pan-cancer miRNA signals. Further study into deconvolution methodologies enabling more accurate quantification of miRNA abundance from purely tumour samples will likely elucidate a clearer picture of miRNA-target interactions. Such deconvolution-based methods would need to ensure that both the expression of miRNA and mRNA are corrected for purity before testing for any such correlations, using methods such

DeconRNASeq[43] or Cibersort[44]. Alternatives to these methods may involve removing samples with low estimates of tumour purity, or removing the miRNA with strong correlations to tumour purity from analysis.

As miRNA are increasingly thought of as potential therapeutic agents, if miRNA are to have effective therapeutic function, a single miRNA may be an ineffective strategy. Rather, a cocktail of miRNA may be needed to modify the tune of the symphony playing within the cancer cell. This has been shown in vitro, with comparisons of multiple miRNA versus a single miRNA targeted highlighting that when multiple miRNA with co-ordinated function are modified, greater phenotypic change is seen, as reported in multiple publications[45,46]. Potentially, for miRNA therapeutics to achieve function, we pose that these may have to be based on a number of miRNA, given to a highly selected group of patients with transcriptomes deemed to be responsive to this network perturbation. Further, by using more than a single miRNA as a therapeutic agent, the off-target effects that have significantly limited development in this field may be mitigated, by buffering for this with other miRNA in off-target tissues[47,48]. In fact, a recent Phase I trial of delivering miR-34a to patients with solid tumours identified acceptable safety overall, but many patients in this trial suffered from off-target immune-related toxicities, such as fever, lymphopenia, and neutropenia, necessitating pre-treatment with dexamethasone[49]. However, we note that this field is in its infancy, and will absolutely require significant further study before even considering clinical translatability, and we offer these ideas as potential avenues to be explored.

In this work we further the knowledge of which miRNA are involved in creating the phenotypes of cancer, across tissue types, to identify miRNA-TSG targets showing exclusive miRNA-mediated suppression. This suggests that a phenomenon similar to that of the previously described BRCA-ness, wherein a miRNA, miR-182, has been shown to repress BRCA and confer sensitivity to PARP inhibitors in a subset of tumours[38], may be at work within many cases, and across multiple tumour suppressor genes. Additionally, recent work has shown how epimutations may result in aberrantly methylated sites that can recapitulate the phenotype of a mutated tumour suppressor such as DNMT3A in leukaemia[50]. This raises the suggestion that there are tumour suppressor genes for which a mutation is not requisite for inactivation, but rather, inactivation is achieved through miRNA-mediated repression or methylation-mediated repression alone. For the TSG we have identified, we have also shown (see Methods section), that the TSG mutations are occurring independently of MYC amplification status, which has been recently identified as an independent regulator of miRNAs. In addition, we show that such MYC amplification status is indeed associated with miRNA expression for the miRNAs found to be negatively associated with each of the TSG in a majority of cases (Supplementary Figure 48, Supplementary Note 12). Further, we have shown that in particular tumours, for PTEN, CDK12 and FAT4, this miRNA or methylation-based suppression happens independently of other gene regulatory factors, such as mutations and copy number changes.

We show how using generally validated, and specifically quality-controlled, gene signatures describing biologically conserved phenotypes can be used to collate large datasets to derive inference about miRNAs, a species whose activation signal has been traditionally hard to detect in smaller cohorts. The ability of this approach to capture tumour biology is highlighted through the identification of tumour suppressor genes showing miRNA-mediated regulation across tumour types, which we have shown have a very strong association to breast cancer molecular subtype. Specifically, this analysis points towards the role of decreased mRNA levels of ARHGEF12, SFRP4 and TGFBR2 in association with the poor-prognosis basal breast cancer subtype[51,52]. Having identified potential negative regulators of these TSG, we show how these miRNAs alone associate with breast cancer subtype, elevated in the basal subtype, capturing a potentially novel biological association.

Finally, the presented methodology may be used in future work encompassing both more specific signatures, as well as larger, more expansive datasets to derive even greater confidence in particular associations. This approach will enable the functional annotation of a greater variety of miRNAs, illuminating their critical role in post-transcriptional gene regulation.

## Methods

**Gene signatures considered**. We consider a wide variety of gene signatures, touching upon many of the hallmarks of cancer, explained in the original and updated work by Hanahan and Weinberg[1,2]. Signatures were selected through a review of MSigDB hallmarks signatures, as well as through a review of the literature, and those used are summarised in Table 1[25]. We note that while many of these signatures were derived for a particular tumour type, we have applied them across many different tumour types, but before doing so, we have performed an evaluation step (sigQC) to ensure that each signature used has suitable properties for application to the datasets under consideration (including normal tissues), in Supplementary Note 1, Supplementary Figures 2–10.

**Datasets considered**. In selecting datasets for this analysis, we initially aimed to select those comprising a comprehensive set of cancer types, with each type represented by a sufficient number of clinical samples, so as to reduce the effects of noise. Thus, we initially began with a consideration of all cancer types represented within the Cancer Genome Atlas datasets (TCGA), and limited based on origin of neoplasm and number of patients for whom miRNA-sequencing was carried out[53]. The RSEM normalised gene expression, mature miRNA normalised expression data, copy number, mutation, and methylation data were accessed from the Firebrowse database at http://www.firebrowse.org. In particular, we considered all cancer types which were epithelial or glandular with respect to histology, and with at least 200 samples with miRNA-sequencing data. These two filters limit the cancers considered to a total of 15 epithelial or glandular neoplasms, comprising a wide variety of cancer types, enabling the strong detection of fundamental biology. Furthermore, among these tumour types, there are 7738 clinical samples, for which 7316 have miRNA-sequencing data. The tumour types, along with their sample counts are presented in Table 2. Details of the number of samples included for each data type are presented in Table 3, and we note that for any analysis presented, any dataset present with fewer than nine samples was excluded from analysis. This restriction excluded the analysis of COAD, OV and UCEC datasets from the analysis of tumour suppressor genes, oncogenes, and exclusivity of regulation.

**miRNA family database**. miRNA ranked across different cancer types were further grouped together by microRNA family, as defined by the targetscan database, implemented in R as the targetscan.Hs.eg.db package, version 0.6.1[54,55].

**Transcriptomic data**. Data were taken from the GDAC Firebrowse TCGA portal provided by the Broad Institute. miRNA datasets used were log2 normalised mature miRNA counts for all cancer types. mRNA datasets used were normalised RSEM genes taken from data through the Illumina HiSeq RNAseq v2 platform. These expression data were then transformed by the transformation $\log_2(x + 1)$, for $x$ as the original expression value, and this was used in all further computation for all cancer types and signatures. Where not otherwise specified, signature scores are taken as the median of log2-transformed expression of all signature genes for each sample. Metabric datasets for normalised miRNA and mRNA expression were taken from the European Genome-Phenome Archive (EGA) under study accession numbers EGAD00010000434 and EGAD00010000438. In all analyses, only miRNA and mRNA expressed at a non-zero level in at least 80% of samples were considered.

**Penalised linear regression**. The aim of the penalised linear regression methodology was to determine those miRNA which most strongly predict (positively or negatively), the gene expression summary score for each signature. With consideration of this, the linear regression was designed such that the model utilised the expression levels of each individual miRNA as a covariate, in order to predict the signature score, taken as the median of the log-transformed expression levels of the signature genes. We note that in order to facilitate direct comparability between distinct signatures and cancer types, we first normalised both the scores and miRNA expression levels to a mean of zero and unit variance. This transformation ensures that the coefficients and their relative values are comparable between cancer types and signatures.

**Table 1 Gene signatures considered and associated hallmarks of cancer**

| Signature name | Reference | Number of genes | Associated hallmarks |
|---|---|---|---|
| Epithelial mesenchymal transition, MSigDB | MSigDB[25] | 200 | Activating invasion and metastasis |
| Invasiveness | Marsan et al., 2014[70] | 16 | Activating invasion and metastasis |
| Oxidative phosphorylation, MSigDB | MSigDB[25] | 200 | Deregulating cellular energetics |
| Reactive oxygen species pathway, MSigDB | MSigDB[25] | 49 | Deregulating cellular energetics |
| G2M checkpoint, MSigDB | MSigDB[25] | 200 | Enabling replicative immortality |
| PI3K-AKT-MTor signalling, MSigDB | MSigDB[25] | 105 | Evading growth suppressors |
| Xenobiotic metabolism, MSigDB | MSigDB[25] | 200 | Evading growth suppressors |
| DNA repair, MSigDB | MSigDB[25] | 150 | Genome instability and mutation, enabling replicative immortality |
| p53 Pathway, MSigDB | MSigDB[25] | 200 | Genome instability and mutation, enabling replicative immortality |
| Hypoxia | Buffa et al.[24] | 51 | Inducing angiogenesis |
| Angiogenesis, MSigDB | MSigDB[25] | 36 | Inducing angiogenesis |
| Hypoxia, MSigDB | MSigDB[25] | 200 | Inducing angiogenesis |
| Angiogenesis, upregulated | Desmedt et al.[71] | 5 | Inducing angiogenesis |
| Angiogenesis | Masiero et al.[69] | 43 | Inducing angiogenesis |
| Apoptosis, MSigDB | MSigDB[25] | 161 | Enabling replicative immortality |
| Apoptosis | Desmedt et al.[71] | 4 | Enabling replicative immortality |
| Proliferation, upregulated | Desmedt et al.[71] | 140 | Sustaining proliferative signalling |
| KRAS signalling, up, MSigDB | MSigDB[25] | 200 | Sustaining proliferative signalling |
| Inflammatory response, MSigDB | MSigDB[25] | 200 | Tumour-promoting inflammation, avoiding immune destruction |
| IL2-STAT5 signalling, MSigDB | MSigDB[25] | 200 | Tumour-promoting inflammation, avoiding immune destruction |
| IL6-JAK-STAT3 signalling, MSigDB | MSigDB[25] | 87 | Tumour-promoting inflammation, avoiding immune destruction |
| TGF$\beta$ signalling, MSigDB | MSigDB[25] | 54 | Tumour-promoting inflammation, avoiding immune destruction |
| TNF$\alpha$ signalling via NF-$\kappa$B, MSigDB | MSigDB[25] | 200 | Tumour-promoting inflammation, avoiding immune destruction |
| Immune invasion, upregulated | Desmedt et al.[71] | 92 | Tumour-promoting inflammation, avoiding immune destruction |

**Table 2 TCGA datasets considered and associated total clinical sample counts**

| Dataset | Abbreviation | Clinical samples |
|---|---|---|
| Breast invasive carcinoma | BRCA | 1098 |
| Ovarian serous cystadenocarcinoma | OV | 602 |
| Lung adenocarcinoma | LUAD | 585 |
| Uterine corpus endometrial carcinoma | UCEC | 560 |
| Kidney renal clear cell carcinoma | KIRC | 537 |
| Head and neck squamous cell carcinoma | HNSC | 528 |
| Lung squamous cell carcinoma | LUSC | 504 |
| Thyroid carcinoma | THCA | 503 |
| Prostate adenocarcinoma | PRAD | 499 |
| Colon adenocarcinoma | COAD | 460 |
| Stomach adenocarcinoma | STAD | 443 |
| Bladder urothelial carcinoma | BLCA | 412 |
| Liver hepatocellular carcinoma | LIHC | 377 |
| Kidney renal papillary cell carcinoma | KIRP | 323 |
| Cervical squamous cell carcinoma and endocervical adenocarcinoma | CESC | 307 |

**Table 3 Counts of samples with miRNA, mRNA, mutation, methylation and copy number data**

| Dataset | mRNA samples | miRNA | mRNA and miRNA | All data |
|---|---|---|---|---|
| BRCA | 782 | 755 | 499 | 324 |
| OV | 307 | 461 | 291 | 0 |
| LUAD | 517 | 452 | 449 | 181 |
| UCEC | 177 | 412 | 174 | 4 |
| KIRC | 534 | 255 | 255 | 121 |
| HNSC | 520 | 486 | 478 | 244 |
| LUSC | 501 | 342 | 342 | 51 |
| THCA | 501 | 502 | 500 | 396 |
| PRAD | 497 | 494 | 493 | 329 |
| COAD | 286 | 221 | 221 | 0 |
| STAD | 415 | 389 | 370 | 230 |
| BLCA | 408 | 409 | 405 | 128 |
| LIHC | 373 | 374 | 369 | 186 |
| KIRP | 291 | 292 | 291 | 148 |
| CESC | 304 | 307 | 304 | 190 |

We used multivariate penalised linear regression, with 10-fold cross validation, as previously described[12] to infer significant relationships between miRNA and gene signatures without overfitting our model. Specifically, first a univariate model filter was applied to the data to select miRNA used for penalised multivariate linear regression. Then, the penalised multivariate linear model with the least predictive error (as assessed on the validating folds) was selected, and coefficients for these miRNA were used for further analysis. All model-fitting, including the initial filtering, was done with 10-fold cross-validation, and was carried out using the penalised package in R, version 0.9–50[56,57]. The initial univariate filter was applied to remove miRNA showing little predictive power from the multivariate linear model, and only those miRNA with $p < 0.2$ significance by F-test in the univariate linear model predicting signature score were considered. This permissive $p$-value was used to assure that the multivariate linear model did not contain artificially stringent associations, as the penalisation procedure also functions as a stringency filter, reducing the false discovery rate. The multivariate linear regression was carried out as a penalised L1/L2 regression to reduce complicating effects of co-correlated miRNAs as predictors of the signature scores. To tune the parameters for the combined L1/L2 regression, a range of values (0, 0.01, 0.1, 1, 10, 100), was tested for the L2 parameter, while in each case the L1 parameter was optimised.

Following computation of all models, the model with the greatest log-likelihood was chosen.

**Rank product analysis.** Once coefficients were obtained for the linear model via the penalised regression approach described earlier, these were collated into matrices with columns defined by cancer type, for each of the gene signatures considered. These coefficients were then fractionally-ranked both from most negative to most positive, and most positive to most negative in value. The rank product statistic, described by Breitling et al. for these fractional ranks was then considered, and the coefficients were ranked in terms of their significance of rank product test statistic, as implemented by the RankProd R package, version 3.6.0[58,59]. This was used to give high-confidence rankings of miRNA associated both positively and negatively with the various signatures considered.

**Validation of miRNA-signature interactions.** In order to ensure reproducibility of the approach used to identify gene signature-associated miRNA, we repeated the linear modelling procedure across the independent Metabric matched miRNA and mRNA microarray dataset of 1293 samples[30]. We mapped each gene signature to corresponding Ensembl IDs, and repeated the combined univariate-multivariate linear modelling approach over all miRNA probes. The miRNA probes identified as positive and negative coefficients were then identified, and mapped to their corresponding mature miRNA ID. The statistical significance of this overlap is shown in Supplementary Figure 12, and was calculated using the Fisher exact test. Nearly all signatures show strong statistical significance, and in the majority of cases not reaching statistical significance, signature applicability to the Metabric dataset may present an issue, as signatures contained a high proportion of genes with low variance, which presents an issue for signature applicability in linear regression, particularly for microarray-based datasets.

**Target analysis.** Targets were aggregated for each miRNA using the miRNAtap database in R, version 1.14.0, as implemented through the Bioconductor targetscan. Hs.eg.db package, version 0.6.1[60]. The default settings of using all 5 possible target databases: DIANA version 5.0[61], Miranda 2010 release[62], PicTar 2005 release[63], TargetScan 7.1[64] and miRDB 5.0[65], with a minimum source number of 2 were used, and the union of all targets found was taken as the set of targets for a given miRNA.

For each of these target-miRNA pairs, the Spearman correlation coefficient was calculated across every cancer type for miRNA versus target mRNA expression, partial to mutation status of the mRNA, and if this value reached statistical significance of $p < 0.05$ (using Spearman correlation asymptotic t-approximation), it was recorded, and otherwise was omitted and recorded as NA. Note that mutational status was reported as a binary variable with a value of 1 for any non-silent, non-intronic mutation, and 0 otherwise. The target-miRNA pairs with at least 5 non-zero entries across cancer types were kept for further analysis, and subsequently were analysed using the rank product statistic, to identify those pairs with consistently negative correlations, across cancer types, with respect to all other hallmarks-miRNA pairs. Partial correlations were done in R using the ppcor package, version 1.1[66].

Furthermore, in the global analysis of all TSG-miRNA pairs, we considered every TSG-miRNA predicted target pair, and again considered the Spearman correlation partial to mutation status, omitting the value as NA if significance $p < 0.05$ (by Spearman correlation asymptotic t-approximation). The rank product statistic was again considered on those pairs with at least 5 non-zero values across cancer types, thereby identifying those TSG-miRNA pairs consistently negatively correlated across cancer types, significantly with respect to all other TSG. Lists of known oncogenes and tumour suppressor genes were taken from the COSMIC database[67]. Because MYC amplification is a possible confounder to the miRNA identified as associated with TSG across cancer types, we checked to ensure that mutation of the 8 TSG identified, across cancer types, does not co-occur significantly with MYC amplification. Of the 96 TSG-cancer type pairs (8 TSG over 12 cancer types), none showed significance in the over-enrichment by a one-sided Fisher exact test for MYC amplification and TSG mutation after correcting for multiple testing.

**Resampling gives significance of TSG among miRNA targets.** To determine the significance of the number of the TSG repressed among the repressed targets of the miRNA identified as signature-associated, we resampled from all miRNA that could possibly selected as signature associated (i.e., those with at least 80% non-zero expression across samples in at least one tumour type), and created 1000 resampled lists of random miRNA of the same length as the number of signature-associated miRNA. Using these lists and the methodology above, miRNA targets were identified, and those miRNA-target correlations (partial to mutation status) consistently negatively ranking compared to all others across tumour types were recorded for each list. Among these repressed targets, we identified the number overlapping with the COSMIC TSG list, and used this to define the empirical distribution of the number of TSG overlapping with the miRNA targets. Then from this distribution, to determine the significance for the 21 TSG overlapping the repressed TSG targets of the signature-associated miRNA, we determined the empirical CDF percentile for the value 21, reported as 0.983, yielding $p = 0.017$ (empirically determined) from this analysis. To ensure that 1000× bootstrap resampling was sufficient, we used the QQ plot for the empirical distribution to ensure close adherence to normality for this distribution.

**Analysis of TSG regulation.** In analysing the regulation of the TSG identified as related to the hallmarks of cancer and potentially amenable to miRNA regulation, we first limited the samples under consideration to those where copy number data, gene expression data, miRNA expression, mutation data, and methylation data were all present. Mutation data was again taken as a binary variable, but as opposed to the partial correlation analysis, mutations were stratified into their reported types (e.g., missense mutations are all grouped together, etc.). That is, the missense mutation variable would only contain a value of 1 if the sample had a missense mutation in the gene of interest, and 0 otherwise. All variables considered in the linear regression were standardised to a mean of 0, and a standard deviation of 1.

L1/2 penalty-based penalised linear regression was then performed, in the same manner as above, for the linear model described in Fig. 3a. Subsequently, coefficients were aggregated across the various cancer types and after the rank product test was applied, those predictors showing statistically consistent positive or negative coefficients were identified. Following this, the autocorrelation of each of these predictor variables was considered, for each of the TSG in each cancer type, as depicted by the heatmap in Fig. 4a.

**Analysis of the exclusivity of gene regulation.** To determine the exclusivity of gene regulation, we calculated the empiric distributions of the variables $\Pi_{\rho k}$ as defined graphically in Fig. 4. These represent the proportion of miRNA-miRNA or miRNA-methylation or methylation probe-methylation probe pairs that show significant positive Spearman co-correlation ($p < 0.05$, by Spearman correlation asymptotic t-approximation). For the bootstrapping analysis, we resampled the datasets, choosing miRNA and methylation probes in the same number as the heatmap in question, and then considered the distributions of the pairwise differences in the variables $\Pi_{\rho k}$. From these distributions for the pairwise differences, we were able to infer the percentile on the empirically constructed CDF that the true case represented, the results of which are depicted in Fig. 4b, showing, for each gene and cancer type, the percentile on the pairwise difference empiric distribution for the observed heatmap.

The calculations for the analysis of TSG regulation and analysis for the exclusivity of gene regulation were repeated for an independent dataset comprising matched mRNA, miRNA, CNV, mutation and methylation data for 93 patients with ovarian cancer, from the OV-AU project from the ICGC data portal[68]. Results of this analysis are highlighted in Supplementary Note 9, Figure 27.

## Data availability

All data used to generate the figures in this paper comes from the GDAC Firebrowse TCGA portal and the EGA as outlined in the methods section above. All code used to generate the data in this paper can be found at https://github.com/andrewdhawan/miRNA_hallmarks_of_cancer/, and cited with: https://doi.org/10.5281/zenodo.1453559.

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

## Acknowledgements

The authors thank Dr. Venkata Manem for helpful discussions. All authors are eternally grateful to the patients and their families who provided tissue to the Cancer Genome Atlas and Metabric projects. The authors are grateful for the financial support provided by Cancer Research UK (F.M.B., A.L.H., A.D.); Clarendon Scholarship (A.D.); Breast Cancer Research Foundation (A.L.H.) and European Research Council (F.M.B.).

## Author contributions

F.M.B. conceived the idea, designed and supervised the study. A.D. and F.M.B. devised the analyses. A.D. wrote code and performed analyses. J.G.S., A.L.H. and F.M.B. supervised the implementation. A.D. wrote the manuscript with contribution from all other authors.

## Additional information

**Competing interests:** The authors declare no competing interests.

