## [Peer Review File · Nature Communications]

Reviewers' comments:

Reviewer #1 (Remarks to the Author):

This is a study on regulation of microRNAs across many cancer sites using data from TCGA and metabarc. There are quite some confusing issues with this manuscript and the methods section is not detailed enough to provide an explanation of how the analysis is done. It is not clear what new knowledge in microRNA molecular biology is being added? The modeling is excellent, but several other similar studies have been conducted including on the TCGA data. The authors have not made a convincing case, that the results here are unique taking into account the state-of-the art. Several studies have mentioned the relationship between microRNAs and tumor suppressor genes and it is not clear, what this study adds to this.

Specific comments:

It is not clear why microRNAs belonging to the same family could not have different functions? What is the rationale that microRNAs in the same family should behave similarly? Have many reports in the past shown this?

It is not clear how the repeat of the analysis in normal tissue is supporting the conclusion that in normal tissue the relationships are not there? This analysis relies on much less normal samples as listed in the methods (20 samples in normal, vs. 200 or more in cancer), can the authors comment this? And argue why the normal analysis is not hampered by this limitation?

Secondly, context dependency between cancer and normal is not that surprising, it would be much more compelling to study the context dependency between different cancer sites, or subtypes within the same cancer (e.g. breast cancer subtypes).

The repetition of the analysis independent of association with hallmark signatures is confusing. Since this second analysis uses all TSGs, why would a TSG-target gene pair identified in the first analysis, have a different result in the second analysis? Otherwise said, shouldn't all results from the first more restricted analysis be a subset of the broader analyses that is independent of the hallmark genes? However, Figure 2 lists 2664 pairs identified in the first analysis and only 31 pairs identified in the second analysis, how is this possible? Since the 2nd analysis is much more broad. The authors mention "Importantly, the conserved miRNA-TSG regulation across cancer types reveals this as a potential new common epigenetic mechanism, alternative to genetic mutations, to achieve functional inhibition of TSGs in cancer." Why is this called an "epigenetic" mechanism?

Analysis depicted in Figure 3a: It is not clear what the iteration over the different methylation probes is in this equation? This is not explained in the methods section to be able to reproduce this analysis. Which methylation probes are considered for each gene? If used genome wide, what is the rationale for using all methylation probes to predict a TSG? Similarly, for the mutation term, what are all the "mut types" that are being considered?

Would you not expect a significantly higher number of microRNAs that have a negative effect on tumor suppressor gene expression in Figure 3b, this is not discussed in the manuscript, and from the figure it seems that both occur evenly likely. Therefore, is the model strategy valid since we do expect a negative relationship between microRNAs and target genes? What is the explanation of the positive relationships?

On the same issue, the authors also mention "These miRNA may be co-expressed for a variety of reasons, such as competitive interactions, repression of repressors of the TSG, or a nearby genomic locus, though penalised regression minimises the effect of co-location because of the inclusion of copy number as a covariate." Wouldn't the "repression of repressors" of the TSG be picked up in the microRNAs that are negatively correlated? Or the model thus does only identify an indirect effect and but how can the direct effect be modeled and discovered?

And it is not clear what this means: "penalised regression minimises the effect of co-location

because of the inclusion of copy number as a covariate" how does penalized regression accomplish this?

Also since it seems that similar numbers of positive relations can be discovered as negative correlations, how can the negative correlations then be taken as more valuable?

It is not clear how the authors showed this "... rather it is subtle changes by a network of miRNAs, interacting with a set of targets in a coordinated manner, that serve to tune the transcriptome to achieve the complex phenotype."

Since the TCGA data set is dominated by the number of breast cancer samples, and the Metabric cohort is also breast cancer, the independent validation might mostly pick up the breast cancer network. What happens when the breast cancer data set is removed from TCGA, is the validation of the miRNA-signature interactions still valid?

Reviewer #2 (Remarks to the Author):

The work is worth publishing, as both the analysis approach and the results are helpful contributions to an analysis area that is far from simple. Below I very briefly summarize the analysis approach, then ask for a number of clarifications, changes, or extensions.

Central points in the method and results are:

1. Page 2, line 51-58. The "repressive signal [of miRNAs] on their targets remains challenging to detect in clinical datasets" [ref 6]. Reference 6 (Buffa et al, PMID: 21737487) reports the penalized regression method used. From 2011, it's cited by over 100 PMC articles. So this statement is rather brief, but probably reasonable for this part of the introduction.
2. Use gene signatures for cancer hallmarks (summarized in Table 3) to get high confidence predictions and associations of miRNAs "and phenotype", where "phenotype" appears to mean: scores for cancer hallmark signatures (that phenotype = a metric from one or more hallmark signatures should be stated very clearly, early on). Page 2, lines 90-93: "we uncover the diverse roles of miRNAs in regulating the hallmarks of cancer. Using signatures seems useful, and it's a helpful contribution to report this set of signatures.
3. Use an R package, sigQC, to check that each signature used could be used with each dataset used. We flag a question around adjacent tissue normals, below.
4. Focus on tumour suppressor genes (TSGs), and on repression of TSGs by miRNA, methylation, mutation, and mutual exclusivity of these alterations across cancers. This is a useful contribution.
5. Use a rank product test to assign a pan-cancer rank to a target gene. This seems reasonable.
6. Discussion, page 11, line 580-583: "we aimed to reduce the complexity of context dependency by selecting interactions that occurred across cancer types". This reasonable approach will deliver insights that apply across cancers, at the potential (or likely) cost of failing to identify insights that are specific to particular cancers.

Major comments

The introduction's discussion around miRNA targeting and its effects should be improved. a) Page 1 line 45: noncanonical miRNA targeting is not mentioned, and the reference [40] is old, 2006. b)

Pages 2 line 51 to line 58: the 2011 reference [6] is the only reference given, and the manuscript fails to note, e.g. all TCGA publications, as well as many others. c) Page 2 line 79 to 82 cites only one reference for ceRNA effects [14] 2016 Molecular [C]ell; there is a large literature on ceRNA issues, with recent publications by Sumazin and Califano being particularly important.

Page 2, line 98 ff: "predicted miRNA-target associations that retain significance across multiple cancer types involve a number of critical tumour suppressor genes and oncogenes". Surprisingly, this fails to cite prior work like the <http://cancerminer.org>'s Jacobsen A. et al, 2013, PMID: 24096364, or any of the 53 publications that cite this reference.

Page 5, line 260-261. Questions:

1. Please state version numbers for all R packages used.
2. From miRNAap package documentation, it's unclear which version of the five predicted binding site resources were used (e.g. Targetscan 7.x?), and how (or whether) predicted targets were ranked (by a predicted 'score' of some sort, Targetscan offers a context++ score or score percentile). The authors may consider that a prediction supported by at least two tools is reliable enough. Would the authors clarify these points.
3. Related to 2, nothing is reported on whether the results were sensitive to filtering to retain higher-confidence targets, and potentially targets for which a 'trained' effect on mRNA levels is expected (e.g. score percentile for TargetScan).

Page 6, lines 346 to 364: The tumour analysis was used on adjacent tissue normals from TCGA, for BRCA, UCEC, HNSC, KIRC, LUAD and BLCA. miRNAs associated with hallmark signature scores were identified. In targets of these positively associated miRNAs, found only 2 of 141 COSMIC tumour suppressors. See also Discussion, page 9, line 528, and page 10 lines 552-564.

Question: Would the authors present results, including sigQC results, that show that cancer hallmark signature scores are meaningful and effective, for this analysis, in samples for adjacent tissue normals.

Page 7, line 432: "negative regulators co-occurred with each other as synergistic repressors". Comment and question: Both co-occurrence and mutual exclusivity should be available in regression results. But would the authors clarify what they did to identify interactions that were 'synergistic'. If they did nothing, so that "synergy" means "co-occurrence", would the authors remove synergy from the text.

Page 8, line 475 ff. Breast cancer subtypes. Eight TSGs were consistently downregulated by miRNAs across cancer types, and expression levels vary across BRCA subtypes. The phrase "levels are inversely associated with... subtype" (line 486-487) is ambiguous, given multiple subtypes; would the authors rewrite this so that it is clearer.

The authors acknowledge ceRNAs, but cite only Denzler 2016 (ref 14, page 11, line 578). Would the authors acknowledge current ceRNA/miRNA work (e.g. by Sumazin and Califano, e.g. PMID: 28558729). The authors appear not to have also considered miRNA expression level, and we'd expect weakly expressed miRNAs to be less likely to influence biology. Would the authors comment on this.

The authors mention tumour purity on page 11, line 585-590, where they point to deconvolution methods to more accurately quantify miRNA abundance.

Questions: As purity would influence inferred expression levels of both miRNAs and target genes, it's not clear whether, for reasonably accurate targeting inferences, across cancers: a) deconvolution would need to be applied to both miRNA -and- mRNA, or b) for a targeting (and gene signature) analysis that depends in part on rank-based correlations, deconvolution would not be needed. Would the authors comment on this, and cite publications that have used such miRNA and mRNA deconvolution/corrections? What alternatives would there be to deconvolution? E.g. remove all samples with purity lower than some threshold? Remove miRNAs and mRNAs whose

abundance was significantly associated with purity?

Discussion, page 11, line 599: would the authors support the statement around "poor therapeutic efficacy" by giving recent references, including references that support treating with multiple miRNAs. The reference [1] given, is from 2007, so is 10 years old.

There are many supplemental figures. While this is potentially helpful, would the authors start these figures by presenting and very carefully documenting several contrasting a TSG cases, so that a reader can easily work with and interpret the many supplemental figures. I'd rate this as 'essential' to the value of the manuscript.

Minor comments

There are surprising errors in references. E.g. At least one reference is incorrectly cited in the References section (e.g. for reference 33, the journal "cell" should be "Cell"). Reference [6], the 'Cancer Research' volume, pages, and year are incomplete. E.g. In citing references, the order of references is odd, currently, starts with [24,25]. Such errors will likely be easily corrected by the citing software (EndNote?).

Page 2, line 50: "and fast-acting effects on cellular phenotype [40]". Why "fast-acting"? Reference [40] is from 2006, so may be reasonable as an older classic reference. But would the authors bring this concept up to date?

Reviewers' comments:

Reviewer 1 (Remarks to the Author):

This is a study on regulation of microRNAs across many cancer sites using data from TCGA and metabarc. There are quite some confusing issues with this manuscript and the methods section is not detailed enough to provide an explanation of how the analysis is done. It is not clear what new knowledge in microRNA molecular biology is being added? The modeling is excellent, but several other similar studies have been conducted including on the TCGA data. The authors have not made a convincing case, that the results here are unique taking into account the state-of-the art. Several studies have mentioned the relationship between microRNAs and tumor suppressor genes and it is not clear, what this study adds to this.

We apologise for the confusion within our manuscript, and thank the reviewer for their careful reading of our manuscript. In the revised version of our manuscript and response below, we have clarified the methods section significantly and we feel that these revisions have very much improved our manuscript. We have also clarified the message of the manuscript and the advances we have shown. In particular, our pan-cancer analysis is the first of its kind to show the exclusivity of the regulation of TSG by miRNA *across cancer types*, in the largest such cohort, lending towards the potential for miRNA-driven tumours. Moreover, our data-driven analysis highlights novel statistical methods taking information from mRNA gene expression signatures, to perform this analysis, yielding high-confidence results. These methods are powerful, and will generalise well to other diseases beyond cancer, where the emerging role of miRNA remains yet to be uncovered, but many well-established mRNA gene signatures exist.

Specific comments:

It is not clear why microRNAs belonging to the same family could not have different functions? What is the rationale that microRNAs in the same family should behave similarly? Have many reports in the past shown this?

We apologise for this lack of clarity in our explanation. The reason for our statement is that members of the same miRNA family share by definition strong common elements of their seed sequence from a common ancestor, and therefore tend to have high redundancy in their targets, for instance, as indicated by Bartel, 2009, Cell (PMID: 19167326), with over 13,000 citations. These have been therefore thought to have in many cases similar functional roles, as explained in Kamanu et al., 2013, Scientific Reports (PMID: 24126940, 39 citations), Lu et al., 2008, PLOS One (PMID: 18923704, 658 citations), or Wang et al., 2010, Bioinformatics (PMID: 20439255, 156 citations) for example. However, we and others have previously reported that this is not always the case; previous specific examples of this in cancer are Korpala et al., Nat Med (PMID:21822286) or van den Beucken et al., Nat Comm (PIMD:25351418). In the present study, once again we confirm this in a more comprehensive analysis covering all miRNA families currently annotated, and we highlight the need to develop a functional classification complementing the family, sequence-based, classification. We have updated the manuscript results section (lines 130-141) and discussion section (lines 313-318) to expand the explanation of the results, including the references above.

It is not clear how the repeat of the analysis in normal tissue is supporting the conclusion that in normal tissue the relationships are not there? This analysis relies on much less normal samples as listed in the methods (20 samples in normal, vs. 200 or more in cancer), can the authors comment this? And argue why the normal analysis is not hampered by this limitation?

The reviewer is absolutely correct in noting that the results of our analysis in normal tissues are supported by fewer samples than tumour samples; a well-known limitation of the dataset we have considered. We felt that despite this limitation, the results of our analysis in normal tissue were indeed worth showing. The implications of these results are that there is less statistical power in this dataset to detect small effects of miRNA, but there are indeed large-scale changes that such an analysis should identify. To present a more

balanced picture of this analysis and its limitations, we have added these discussions of limitation into the main text on lines 209-212. Moreover, for full clarity of the quality of the application of the gene signatures on the normal datasets, we have included these *sigQC* results in the supplementary information as well showing that their application is comparable to tumour datasets (Figs S2-S10). Further, to further investigate whether we could identify differences between the miRNA identified as hallmarks-associated in cancer samples versus normal samples, we considered the similarity between the identified miRNA for *only* the tumour samples with matched normal samples. We describe this analysis and its result in Supplementary Section 5c, showing that across nearly every signature considered, even with this reduced set of samples, we are able to identify differences between normal and tumour samples.

Secondly, context dependency between cancer and normal is not that surprising, it would be much more compelling to study the context dependency between different cancer sites, or subtypes within the same cancer (e.g. breast cancer subtypes).

While context dependency between cancer and normal is not necessarily surprising, we do feel that it is an under-reported result in the literature, especially when shown as analysis of both miRNAs and targets, and a result through a large, data-driven study of clinical specimens. This has not been shown in the literature before, to our knowledge. We agree that context dependency between different cancer types or breast cancer subtypes are compelling results, but we must also take measures to ensure that results of context dependency are not conflated with noisy results arising from small population sizes, especially for miRNA, where the effect can be difficult to detect in clinical samples. In fact, we did test the differences between breast cancer subtypes, but the models describing this were clearly overfit due to small sample size. Moreover, when considering a comparison between different cancer types, we desired consistent behaviour across multiple models for a given miRNA predictor, so that differences could more confidently be assigned to context dependency, as opposed to model fitting. That is, showing (perhaps less compellingly) that there are key differences between tumour and normal, as opposed to differences between specific cancers was a decision we made to ensure robustness of our results. We have updated the manuscript to discuss this issue, and the potential for future studies with larger cohorts of patients to detect context dependency between tumour subtypes or different tumour types on lines 324-328.

The repetition of the analysis independent of association with hallmark signatures is confusing. Since this second analysis uses all TSGs, why would a TSG-target gene pair identified in the first analysis, have a different result in the second analysis? Otherwise said, shouldn't all results from the first more restricted analysis be a subset of the broader analyses that is independent of the hallmark genes? However, Figure 2 lists 2664 pairs identified in the first analysis and only 31 pairs identified in the second analysis, how is this possible? Since the 2nd analysis is much more broad.

We apologise for the confusion. To reduce our false discovery rate of miRNA-mRNA target interactions with respect to tumour suppressor genes, we felt it necessary to use a two-pronged validation strategy. That is, we sought *only* to include those interactions which showed very strong evidence for miRNA-TSG repression across cancer types. We clarify the approach we have used as follows. Our first step was to consider all hallmarks-associated miRNA and predicted mRNA targets (including miRNA-TSG interactions, as predicted by target prediction algorithms). Analysing these correlations across cancer types, we identified 17895 pairs for which there were enough significant values to consider the rank product across cancer types, which is where 2664 pairs were identified as significant. The second approach considered every predicted miRNA-TSG pair, *individually* for each TSG, and considered the correlation partial to mutation status. That is, the second part asked the question, 'given a TSG, and all of the miRNA predicted to repress it, which are the most consistently negative across cancer types?' This approach, by then considering each TSG independently, yielded the strongest candidate miRNA repressing each TSG, which we aggregated into the list of 31 pairs. We have added clarification for this aspect of our manuscript in the main text on lines 184-186; we thank the reviewer for pointing this out and apologise for this confusion. We have also added Supplementary Figure S1 describing these methods in a flow-diagram. We thank the reviewer for this point and we hope that all these changes together add the necessary clarity.

The authors mention “Importantly, the conserved miRNA-TSG regulation across cancer types reveals this as a potential new common epigenetic mechanism, alternative to genetic mutations, to achieve functional inhibition of TSGs in cancer.” Why is this called an “epigenetic” mechanism?

We agree this terminology was confusing. We had used the term epigenetic by its broad definition, but in the interest of clarity in our manuscript, we have removed the word ‘epigenetic’ to describe miRNA-mRNA repressive regulation. We thank the reviewer for raising this point of confusion.

Analysis depicted in Figure 3a: It is not clear what the iteration over the different methylation probes is in this equation? This is not explained in the methods section to be able to reproduce this analysis. Which methylation probes are considered for each gene? If used genome wide, what is the rationale for using all methylation probes to predict a TSG? Similarly, for the mutation term, what are all the “mut types” that are being considered?

To add clarity to this analysis, we have listed every mutation type (as reported in the Firebrowse TCGA datasets) and each methylation probe used for the analysis of each TSG in the supplementary information of our manuscript (Supplementary Tables S1 and S2).

Would you not expect a significantly higher number of microRNAs that have a negative effect on tumor suppressor gene expression in Figure 3b, this is not discussed in the manuscript, and from the figure it seems that both occur evenly likely. Therefore, is the model strategy valid since we do expect a negative relationship between microRNAs and target genes? What is the explanation of the positive relationships?

In general we might expect a significantly higher number of miRNA negatively related to TSG expression, if the miRNA were *a priori* expected to target the TSG of interest. However, in this analysis, we have instead considered every miRNA (not just those predicted to target a given TSG), in our linear model. Our rationale for this in our modelling approach was to ensure that we did not miss any non-predicted repressive actions of miRNA and TSG, and to potentially detect novel associations that might have been missed by target prediction algorithms. Furthermore, we were wary of inducing bias into this analysis by including only predicted miRNA interactions. As such, it is not unexpected that many miRNAs associate positively with TSG expression, as many of these are potentially co-expressed. This is explicable, as some of these miRNA may be on the same genomic locus as the TSG, controlled by the same enhancer region, or related positively to TSG expression by ceRNA effects. To clarify this to the reader of our manuscript, we have improved this discussion in our manuscript on lines 234-244.

On the same issue, the authors also mention “These miRNA may be co-expressed for a variety of reasons, such as competitive interactions, repression of repressors of the TSG, or a nearby genomic locus, though penalised regression minimises the effect of co-location because of the inclusion of copy number as a covariate.” Wouldn’t the “repression of repressors” of the TSG be picked up in the microRNAs that are negatively correlated? Or the model thus does only identify an indirect effect and but how can the direct effect be modeled and discovered?

This wording may be confusing for the reader, and we thank the reviewer for flagging it. Repression of the repressors of the TSG may cause miRNA to be co-correlated, in that if the repressors are mRNA themselves, they may be co-expressed with the TSG. For example if miRNA A targets mRNA X which represses the TSG Y, then miRNA A and the TSG Y will appear to be co-expressed in our model, because as A increases, X decreases, which results in a concomitant increase in Y. The repression of repressors does not refer to other miRNA, and we recognise that this may be the nature of the confusion. This is, in fact, a strong introduction to the discussion of competing endogenous RNA (ceRNA); the effects of which are an area of current research. With guidance from the second reviewer as well, we have revised our discussion of the ceRNA effect, including more relevant current publications. To answer the second part of this question, we note that we take any negative effect that is preserved across cancer types in our model as a direct effect of a miRNA on a mRNA, as this is the simplest explanation for this finding, but submit that these findings may require validation

experimentally; but this is the best that can be done from our data-driven approach. We have revised the text to include this discussion, on lines 234-244.

And it is not clear what this means: “penalised regression minimises the effect of co-location because of the inclusion of copy number as a covariate” how does penalized regression accomplish this?

We apologise for this lack of clarity in our methods. Penalised linear regression applies two penalty terms in the fitting of the linear model, a selection term (L1 penalty) increasing interpretability of the model and a regularization term (L2 penalty) avoiding over-fitting. The latter enables the model to achieve reliable fits in the case of correlated covariates. In our case, the covariates for copy number and expression of miRNA on the same genomic locus are highly correlated, and the result of including a L2 penalty is a reliable redistribution of the effect between them. As such, the effect of both co-variables is more fairly addressed in the model, and the importance of one term is not inflated. We have clarified this discussion in the manuscript for the reader, on lines 234-244.

Also since it seems that similar numbers of positive relations can be discovered as negative correlations, how can the negative correlations then be taken as more valuable?

This is an important point raised by the reviewer. As clarified above, this part of the analysis considers both predicted targets and other genes. Thus, by expanding the field of interactions that we consider to genes that are not predicted targets, we do expect to identify many positive relationships. The negative interactions are taken as more valuable in this case, because of the underlying biology that they potentially represent. Negative correlations potentially represent the inhibitory actions of miRNA on the TSG, which are the dominant predicted role of miRNA. To clarify this in the manuscript, we have added the following text to lines 248-249.

It is not clear how the authors showed this “... rather it is subtle changes by a network of miRNAs, interacting with a set of targets in a coordinated manner, that serve to tune the transcriptome to achieve the complex phenotype.”

We agree that this statement may be misleading to the reader. As such we have revised it to read as follows: “rather it appears to be that changes from an expressed network of miRNAs, interacting with a set of targets in a coordinated manner, serve to tune the transcriptome to achieve the complex phenotypes associated with cancer.” on lines 331-333.

Since the TCGA data set is dominated by the number of breast cancer samples, and the Metabric cohort is also breast cancer, the independent validation might mostly pick up the breast cancer network. What happens when the breast cancer data set is removed from TCGA, is the validation of the miRNA-signature interactions still valid?

This is an important question raised by the reviewer. In effect, by aggregating statistics across cancer types using the rank product, we have applied the same weighting to all cancer types. However, because the sizes of the breast cancer datasets, there are likely associations that could be detected in these datasets that could not be detected in others. However, because these associations would only have been detected in breast cancer, and the magnitude of the associations would have been small, they would not have remained after the rank product analysis. As such, we expect our results to be robust to removing the breast cancer dataset from TCGA. To highlight this, we have shown that the overlap is greater than 75% in all cases of signatures, when considering positively and negatively associated miRNA. We have added this information into Supplementary Section S5b, and reported this in the main text on lines 123-127.

Reviewer 2 (Remarks to the Author):

The work is worth publishing, as both the analysis approach and the results are helpful contributions to an analysis area that is far from simple. Below I very briefly summarize the analysis approach, then ask for a number of clarifications, changes, or extensions.

Central points in the method and results are:

1. Page 2, line 51-58. The “repressive signal [of miRNAs] on their targets remains challenging to detect in clinical datasets” [ref 6]. Reference 6 (Buffa et al, PMID: 21737487) reports the penalized regression method used. From 2011, it’s cited by over 100 PMC articles. So this statement is rather brief, but probably reasonable for this part of the introduction.

2. Use gene signatures for cancer hallmarks (summarized in Table 3) to get high confidence predictions and associations of miRNAs “and phenotype”, where “phenotype” appears to mean: scores for cancer hallmark signatures (that phenotype = a metric from one or more hallmark signatures should be stated very clearly, early on). Page 2, lines 90-93: “we uncover the diverse roles of miRNAs in regulating the hallmarks of cancer. Using signatures seems useful, and it’s a helpful contribution to report this set of signatures.

3. Use an R package, sigQC, to check that each signature used could be used with each dataset used. We flag a question around adjacent tissue normals, below.

4. Focus on tumour suppressor genes (TSGs), and on repression of TSGs by miRNA, methylation, mutation, and mutual exclusivity of these alterations across cancers. This is a useful contribution.

5. Use a rank product test to assign a pan-cancer rank to a target gene. This seems reasonable.

6. Discussion, page 11, line 580-583: “we aimed to reduce the complexity of context dependency by selecting interactions that occurred across cancer types”. This reasonable approach will deliver insights that apply across cancers, at the potential (or likely) cost of failing to identify insights that are specific to particular cancers.

We thank the reviewer for the very positive comments, and thank them for their thorough review of our manuscript. The revisions as inspired from the comments below have certainly strengthened our analysis and presentation, and we have outlined our responses below.

Major comments

The introduction’s discussion around miRNA targeting and its effects should be improved. a) Page 1 line 45: noncanonical miRNA targeting is not mentioned, and the reference [40] is old, 2006.

We have worked to improve our introduction in the revised version of our manuscript. Following the suggestion of the reviewer, we have added further discussion about miRNA targeting, target prediction, and non-canonical miRNA targeting. The updated text describing these additions can be found on lines 21-38 of the revised manuscript. We have added the following more current references: Loeb et al., 2012, Molecular Cell (PMID: 23142080); Betel et al., 2010, Genome Biology (PMID: 20799968); Cloney, 2016, Nature Reviews Genetics (PMID: 27795565); Hausser and Zavolan, 2014, Nature Reviews Genetics (PMID: 25022902); Agarwal et al., 2015, eLife (PMID: 26267216); and Jonas and Izaurralde, 2015, Nature Reviews Genetics (PMID: 26077373).

b) Pages 2 line 51 to line 58: the 2011 reference [6] is the only reference given, and the manuscript fails to note, e.g. all TCGA publications, as well as many others.

We thank the reviewer for pointing out this deficiency in our work, and we have revised the introduction to discuss the work done by the TCGA and others, such as Jacobsen et al. Revised text can be found on lines

28-38. We have added the following references for a selection of TCGA projects involving miRNA study: Dvinge et al., 2013, *Nature* (PMID: 23644459); Brennan et al., 2013, *Cell* (PMID: 24120142); Davis et al., 2014, *Cancer Cell* (PMID: 25155756); and Bolouri et al., 2018, *Nature Medicine* (PMID: 29227476).

c) Page 2 line 79 to 82 cites only one reference for ceRNA effects [14] 2016 Molecular [C]ell; there is a large literature on ceRNA issues, with recent publications by Sumazin and Califano being particularly important.

Again, we apologise for this oversight. We have added the discussion of these key references for ceRNA in the introduction on lines 51-53. As suggested, we have added Chiu et al., 2017, *BMC Genomics* (PMID: 28558729); and have also added reference to the work done by Xu et al., 2016, *Nucleic Acids Research* (PMID: 27365046) in this area.

Page 2, line 98 ff: "predicted miRNA-target associations that retain significance across multiple cancer types involve a number of critical tumour suppressor genes and oncogenes". Surprisingly, this fails to cite prior work like the <http://cancerminer.org>'s Jacobsen A. et al, 2013, PMID: 24096364, or any of the 53 publications that cite this reference.

We thank the reviewer for pointing out this significant deficiency in our work. We have added this key citation, and note in the introduction what Jacobsen et al. had achieved, and that our work extends this study further, as reported on lines 34-38.

Page 5, line 260-261. Questions:

1. Please state version numbers for all R packages used.

We apologise for this oversight and have added version numbers for all R packages we have used.

2. From miRNAAtap package documentation, it's unclear which version of the five predicted binding site resources were used (e.g. Targetscan 7.x?), and how (or whether) predicted targets were ranked (by a predicted 'score' of some sort, Targetscan offers a context++ score or score percentile). The authors may consider that a prediction supported by at least two tools is reliable enough. Would the authors clarify these points.

After reaching out to the maintainer of the miRNAAtap package, we have clarified the version of the miRNAAtap tools that were used are as follows: TargetScan - 7.1 (2016), DIANA - 5.0 (2013), Miranda - August 2010 release, PicTar - 2005 publication, and miRDB - 5.0 (2014). These predictions were indeed ranked by their predicted scores (e.g. context++ score in Targetscan), and these ranks were used by miRNAAtap to run a rank product analysis to identify those targets with significant high ranks across all target prediction algorithms. We have added this information into the manuscript on lines 484-485 to clarify.

3. Related to 2, nothing is reported on whether the results were sensitive to filtering to retain higher-confidence targets, and potentially targets for which a 'trained' effect on mRNA levels is expected (e.g. score percentile for TargetScan).

This is an important concern by the reviewer. While we have not repeated our analysis further to include greater stringency in the target selection (e.g. requiring more than two sources for a target to be considered), we note that our analysis methodology itself is designed to obtain high-confidence targets. At every step of our analysis, we have worked to develop methods to include as many possible miRNA/mRNA targets, without a high rate of false discovery. For instance, including a minimum of two sources ensures we are including a reasonable number of miRNA, while not restricting to those which are entirely predicted by the commonalities of each prediction algorithm. With this potentially more comprehensive list, we subsequently seek to test which of these miRNA/target pairs themselves show repressive potential using correlations in the data, and then further refine this list by using a rank product statistic. In essence, for this methodology to work, we require a reasonably wide and comprehensive list of potential targets initially, as those which are not reasonable choices will be removed by further analysis, especially in our rank product.

Page 6, lines 346 to 364: *The tumour analysis was used on adjacent tissue normals from TCGA, for BRCA, UCEC, HNSC, KIRC, LUAD and BLCA. miRNAs associated with hallmark signature scores were identified. In targets of these positively associated miRNAs, found only 2 of 141 COSMIC tumour suppressors. See also Discussion, page 9, line 528, and page 10 lines 552-564.*

Question: Would the authors present results, including sigQC results, that show that cancer hallmark signature scores are meaningful and effective, for this analysis, in samples for adjacent tissue normals.

This is an excellent suggestion, and we have included these results in the Supplementary Figures S2-S10, showing that indeed, these signatures are applicable in terms of their statistical behavior to the adjacent tissue normal datasets. We have also referenced these and addressed this issue in the main text more clearly on lines 209-212.

Page 7, line 432: *“negative regulators co-occurred with each other as synergistic repressors”. Comment and question: Both co-occurrence and mutual exclusivity should be available in regression results. But would the authors clarify what they did to identify interactions that were ‘synergistic’. If they did nothing, so that “synergy” means “co-occurrence”, would the authors remove synergy from the text.*

Thank you for pointing this out. We apologise for the confusion regarding our wording, as recommended, we have removed the term synergy so as not to confuse the analysis that was done.

Page 8, line 475 ff. *Breast cancer subtypes. Eight TSGs were consistently downregulated by miRNAs across cancer types, and expression levels vary across BRCA subtypes. The phrase “levels are inversely associated with... subtype” (line 486-487) is ambiguous, given multiple subtypes; would the authors rewrite this so that it is clearer.*

We thank the reviewer for their careful reading of the manuscript and have corrected the wording to be less ambiguous and directly state how miRNA and mRNA levels vary with tumour subtype, on lines 280-282.

The authors acknowledge ceRNAs, but cite only Denzler 2016 (ref 14, page 11, line 578). Would the authors acknowledge current ceRNA/miRNA work (e.g. by Sumazin and Califano, e.g. PMID: 28558729). The authors appear not to have also considered miRNA expression level, and we’d expect weakly expressed miRNAs to be less likely to influence biology. Would the authors comment on this.

We thank the reviewer for pointing out this deficiency in our manuscript, and as such have added the recommended manuscript, and other manuscripts as follows: Xu et al., 2016, Nucleic Acids Research (PMID: 27365046); Denzler et al., 2014, Molecular Cell (PMID: 24793693); Smillie et al., 2018, Critical Reviews in Biochemistry and Molecular Biology (PMID: 29569941); and Zarringhalam, 2017, Scientific Reports (PMID: 28798471), on lines 331-340. We also clarify that indeed, we have considered miRNA expression level in our analysis by using a threshold ensuring that at least 80% of samples express a given miRNA at a non-zero level, as indicated on lines 434-435. We have added a comment in the manuscript describing that miRNA that are poorly expressed are less likely to affect biology.

The authors mention tumour purity on page 11, line 585-590, where they point to deconvolution methods to more accurately quantify miRNA abundance.

Questions: As purity would influence inferred expression levels of both miRNAs and target genes, it’s not clear whether, for reasonably accurate targeting inferences, across cancers: a) deconvolution would need to be applied to both miRNA -and- mRNA, or b) for a targeting (and gene signature) analysis that depends in part on rank-based correlations, deconvolution would not be needed. Would the authors comment on this, and cite publications that have used such miRNA and mRNA deconvolution/corrections? What alternatives would there be to deconvolution? E.g. remove all samples with purity lower than some threshold? Remove miRNAs and mRNAs whose abundance was significantly associated with purity?

This is a very thought-provoking question. Given that we are developing methods to infer miRNA-mRNA relationships using gene signature information, it is critical that the expression of both datasets is corrected for tumour purity; potentially by deconvolution methods if these are available. We have added this important consideration into the manuscript, on lines 346-349. Regarding point (b), we agree that given that we are looking at rank-based correlations, these are less likely to be affected by deconvolution methods, but to our knowledge this has not been rigorously shown (i.e. by a head-to-head comparison of rank-based correlation in convoluted and deconvoluted datasets). Moreover, to our knowledge, there are no published deconvolution methods that have been designed for miRNA datasets, but these certainly would be a very important potential future contribution to the literature. We have added references to the manuscript that describe findings associated with mRNA deconvolution (Gong et al., 2013, *Bioinformatics* (PMID: 23428642), and Gentles et al., 2015, *Nature Medicine* (PMID: 26193342)), but note that there are at present no works we could identify that used this to infer miRNA function. We have also added a discussion of alternatives to deconvolution, and as the reviewer very astutely suggests, discuss the potential of using tumour purity cutoffs, or removing miRNA/mRNA with strong associations to tumour purity.

Discussion, page 11, line 599: would the authors support the statement around “poor therapeutic efficacy” by giving recent references, including references that support treating with multiple miRNAs. The reference [1] given, is from 2007, so is 10 years old.

Absolutely - we have added the following more recent references to support our statement: Yilmazel, 2014, *BMC Bioinformatics* (PMID: 24934636), regarding off target effects of miRNA; Beg, 2017, *Investigational New Drugs* (PMID: 27917453), describing a Phase I trial of synthetic miR-34a; Rothschild, 2014, *Molecular and Cellular Therapies* (PMID: 26056576) and Rupaimoole, 2017, *Nature Reviews Drug Discovery* (PMID: 28209991) for current reviews of the field; Lu, 2009, *Immunity* (PMID: 26163372), Yang, 2015, *Clinical Cancer Research* (PMID: 25724519), Lee, 2015, *Oncotarget* (PMID: 26375442), Brognara, 2016, *International Journal of Oncology* (PMID: 26708164), and Zhang, 2009, *International Journal of Oncology* (PMID: 19424584) describing the concept of targeting multiple miRNA to change phenotype. With respect to our statement of poor therapeutic efficacy, we have re-worded this to be more fair, given the lack of current clinical trials, and have removed this. The modified text can be found on lines 352-364.

There are many supplemental figures. While this is potentially helpful, would the authors start these figures by presenting and very carefully documenting several contrasting a TSG cases, so that a reader can easily work with and interpret the many supplemental figures. I'd rate this as 'essential' to the value of the manuscript.

We thank the reviewer for this helpful comment - we have added significantly to Supplementary Section S8, describing the example case of our approach to how we determined and studied the exclusivity of regulation of the TSG we identified as showing evidence for miRNA regulation. We have written this discussion contrasting the results seen for the case of ACVR2A in stomach adenocarcinoma, which shows little evidence for exclusivity in its regulation between miRNA, methylation, and mutation; and the case of PTEN in bladder cancer, which shows strong evidence of exclusivity. We have added a supplementary figure as well to help depict this flow graphically, and thank the reviewer for this suggestion; we feel that it has certainly enhanced the readability of our manuscript, and the comprehensiveness of our methods.

Minor comments

There are surprising errors in references. E.g. At least one reference is incorrectly cited in the References section (e.g. for reference 33, the journal “cell” should be “Cell”). Reference [6], the ‘Cancer Research’ volume, pages, and year are incomplete. E.g. In citing references, the order of references is odd, currently, starts with [24,25]. Such errors will likely be easily corrected by the citing software (EndNote?).

We apologise for these errors in our referencing. We have corrected all references in the revised version of our manuscript.

Page 2, line 50: “and fast-acting effects on cellular phenotype [40]”. Why “fast-acting”? Reference [40] is from 2006, so may be reasonable as an older classic reference. But would the authors bring this concept up to date?

We thank the reviewer again for their thorough reading of our manuscript - it has definitely improved our work. We agree that the term fast-acting may be misleading, and we have removed it. In response to the question of reference used, we have added the following (more recent, and relevant) reference: Bracken, 2016, Nature Reviews Genetics (PMID: 27795564) to highlight the network aspect of the miRNA-target interaction across the transcriptome.

Reviewers' comments:

Reviewer #1 (Remarks to the Author):

The authors have extensively covered almost all issues from the previous review. One aspect remains that is argued both ways by the authors:

- The response to the question regarding the normal vs. the comparison between subsets of cancers is conflicting with the next item (within cancer subtype comparisons). The analysis added by the authors regarding the limited size of normal is very strong and shows that the results are definitely not affected by having less normal. The authors then conclude that the analysis is still strong even with only 20 (tumor) samples. However, the arguments in the next part are the opposite and now the authors argue that that analysis fails because there are not enough samples. Note that in breast cancer the studied cohorts should contain at least 20 samples per subtype. Also it has been reported several times that the main breast cancer subtypes, can be thought of as very different diseases (e.g. basal vs. non-basal), therefore the fact that the authors cannot pick up differential regulation of microRNAs between different breast cancer subtypes is disappointing.

Minor comment:

- The SiqQC radar plots might not be familiar to most readers, more details can be added to the methods/legends/captions or anywhere the authors deem it appropriate to explain at least in one spot the details of how to read such a plot.

Reviewer #2 (Remarks to the Author):

The authors have responded carefully to both reviewers. The manuscript is worth publishing, largely because of how well it describes issues, methods, and results.

I've flagged two issues that the authors should still address.

The paragraph in lines 352 to 366, which speaks of personalized treatment with a single miRNA vs. multiple miRNAs, should be adjusted to be more cautious.

From the rebuttal document.

3. Related to 2, nothing is reported on whether the results were sensitive to filtering to retain higher-confidence targets, and potentially targets for which a 'trained' effect on mRNA levels is expected (e.g. score percentile for TargetScan).

[Author response] This is an important concern by the reviewer. While we have not repeated our analysis further to include greater stringency in the target selection (e.g. requiring more than two sources for a target to be considered), we note that our analysis methodology itself is designed to obtain high-confidence targets. At every step of our analysis, we have worked to develop methods to include as many possible miRNA/mRNA targets, without a high rate of false discovery. For instance, including a minimum of two sources ensures we are including a reasonable number of

miRNA, while not restricting to those which are entirely predicted by the commonalities of each prediction algorithm. With this potentially more comprehensive list, we subsequently seek to test which of these miRNA/target pairs themselves show repressive potential using correlations in the data, and then further refine this list by using a rank product statistic. In essence, for this methodology to work, we require a reasonably wide and comprehensive list of potential targets initially, as those which are not reasonable choices will be removed by further analysis, especially in our rank product.

In the Discussion, would the authors include a brief summary of the question and their response.

Minor issues

Spelling. Line 74: "In summery,..."

Line 124: "Supplementary Figure ??"

Line 184: "in Figure ??"

Line 207: Figures ??-??

Line 301: Manuscript needs careful proof-reading. E.g. "Thus, this analysis represents a novel and powerful assessment of the intricacy of miRNA regulation of phenotypes, particularly important in the context of a complex disease such as cancer." Suggest: "..., which is particularly important".

Line 335: "repressable" should probably be "repressible"

Reviewer 1:

The authors have extensively covered almost all issues from the previous review. One aspect remains that is argued both ways by the authors:

- The response to the question regarding the normal vs. the comparison between subsets of cancers is conflicting with the next item (within cancer subtype comparisons). The analysis added by the authors regarding the limited size of normal is very strong and shows that the results are definitely not affected by having less normal. The authors then conclude that the analysis is still strong even with only 20 (tumor) samples. However, the arguments in the next part are the opposite and now the authors argue that that analysis fails because there are not enough samples. Note that in breast cancer the studied cohorts should contain at least 20 samples per subtype. Also it has been reported several times that the main breast cancer subtypes, can be thought of as very different diseases (e.g. basal vs. non-basal), therefore the fact that the authors cannot pick up differential regulation of microRNAs between different breast cancer subtypes is disappointing.

We apologise for the lack of clarity in our response, and we agree that identifying differential regulation between breast cancer subtypes is a crucially important aspect. In our previous response, we did highlight the differences in sample size, but we did not discuss in detail the difference in the number of independent datasets. Specifically, the results shown have been obtained using normal/tumour datasets with at least 20 samples (across at least 5 datasets), with the rank product statistic. In the comparison of breast cancer subtypes, the samples all come from the same dataset, and so no filtering based on the rank product is used when identifying miRNA. It is this loss in statistical power (that comes from not using the rank product statistic) that results in the potential overfitting of the models to these datasets. Despite this statistical limitation, we were able to successfully use our method to identify a signal of context dependency in the miRNA involved with the hallmarks between the two subtypes. We have added these new results to Supplementary Section 5d, incorporated Figure S15, and added discussion of this on lines 220-227 of the manuscript. Again, we thank the reviewer for this excellent suggestion and direction in further showing the abilities of our approach.

Minor comment:

- The SigQC radar plots might not be familiar to most readers, more details can be added to the methods/legends/captions or anywhere the authors deem it appropriate to explain at least in one spot the details of how to read such a plot.

We thank the reviewer for this pertinent suggestion. We have added these details describing the interpretation of the metrics of the sigQC radar plots to the manuscript on lines 19-42 in Supplementary section S2.

Reviewer 2:

The authors have responded carefully to both reviewers. The manuscript is worth publishing, largely because of how well it describes issues, methods, and results.

We thank the reviewer for their positive feedback on our manuscript. The revisions from both reviewers have been incredibly helpful in producing our manuscript and increasing its scientific quality.

I've flagged two issues that the authors should still address.

The paragraph in lines 352 to 366, which speaks of personalized treatment with a single miRNA vs. multiple miRNAs, should be adjusted to be more cautious.

We have updated the text in this section (lines 373-387) to be more cautious as the reviewer suggests. Thank you for pointing this out.

From the rebuttal document. 3. Related to 2, nothing is reported on whether the results were sensitive to filtering to retain higher-confidence targets, and potentially targets for which a 'trained' effect on mRNA levels is expected (e.g. score percentile for TargetScan).

[Author response] This is an important concern by the reviewer. While we have not repeated our analysis further to include greater stringency in the target selection (e.g. requiring more than two sources for a target to be considered), we note that our analysis methodology itself is designed to obtain high-confidence targets. At every step of our analysis, we have worked to develop methods to include as many possible miRNA/mRNA targets, without a high rate of false discovery. For instance, including a minimum of two sources ensures we are including a reasonable number of miRNA, while not restricting to those which are entirely predicted by the commonalities of each prediction algorithm. With this potentially more comprehensive list, we subsequently seek to test which of these miRNA/target pairs themselves show repressive potential using correlations in the data, and then further refine this list by using a rank product statistic. In essence, for this methodology to work, we require a reasonably wide and comprehensive list of potential targets initially, as those which are not reasonable choices will be removed by further analysis, especially in our rank product.

In the Discussion, would the authors include a brief summary of the question and their response.

We have added this important question and its response to the discussion section of our manuscript on lines 338-350.

Minor issues

- *Spelling. Line 74: "In summery,..."*
- *Line 124: "Supplementary Figure ??"*
- *Line 184: "in Figure ??"*
- *Line 207: Figures ??-??*
- *Line 301: Manuscript needs careful proof-reading. E.g. "Thus, this analysis represents a novel and powerful assessment of the intricacy of miRNA regulation of phenotypes, particularly important in the context of a complex disease such as cancer." Suggest: "..., which is particularly important".*
- *Line 335: "repressable" should probably be "repressible"*

We thank the reviewer again for their careful reading of our manuscript - we have proofread our manuscript again, and have addressed each of these minor issues.

REVIEWERS' COMMENTS:

Reviewer #1 (Remarks to the Author):

All comments have been addressed.